# Immunoprophylactic and immunotherapeutic control of hormone receptor-positive breast cancer

Aitziber Buqué [1,2], Norma Bloy[1,2], Maria Perez-Lanzón[2,3,4], Kristina Iribarren[2,4], Juliette Humeau[2,3,4], Jonathan G. Pol[2,4], Sarah Levesque[2,3,4], Laura Mondragon[2,4], Takahiro Yamazaki [1], Ai Sato [1], Fernando Aranda [2,4], Sylvère Durand[2,4], Alexandre Boissonnas[5], Jitka Fucikova[6,7], Laura Senovilla[2], David Enot[2,4], Michal Hensler[6], Margerie Kremer[2,4], Gautier Stoll [2,4], Yang Hu [8], Chiara Massa[9], Silvia C. Formenti [1,10], Barbara Seliger[9], Olivier Elemento[8,10], Radek Spisek[6,7], Fabrice André [11], Laurence Zitvogel[3,11,12,13], Suzette Delaloge [14], Guido Kroemer [2,4,15,16,17,20✉] & Lorenzo Galluzzi [1,8,10,18,19,20✉]

Hormone receptor (HR)$^+$ breast cancer (BC) causes most BC-related deaths, calling for improved therapeutic approaches. Despite expectations, immune checkpoint blockers (ICBs) are poorly active in patients with HR$^+$ BC, in part reflecting the lack of preclinical models that recapitulate disease progression in immunocompetent hosts. We demonstrate that mammary tumors driven by medroxyprogesterone acetate (M) and 7,12-dimethylbenz[a] anthracene (D) recapitulate several key features of human luminal B HR$^+$HER2$^-$ BC, including limited immune infiltration and poor sensitivity to ICBs. M/D-driven oncogenesis is accelerated by immune defects, demonstrating that M/D-driven tumors are under immunosurveillance. Safe nutritional measures including nicotinamide (NAM) supplementation efficiently delay M/D-driven oncogenesis by reactivating immunosurveillance. NAM also mediates immunotherapeutic effects against established M/D-driven and transplantable BC, largely reflecting increased type I interferon secretion by malignant cells and direct stimulation of immune effector cells. Our findings identify NAM as a potential strategy for the prevention and treatment of HR$^+$ BC.

[1] Department of Radiation Oncology, Weill Cornell Medical College, New York, NY, USA. [2] Equipe labellisée par la Ligue contre le cancer, Centre de Recherche des Cordeliers, INSERM U1138, Université de Paris, Sorbonne Université, Paris, France. [3] Faculté de Médecine, Université de Paris Sud, Paris-Saclay, Le Kremlin-Bicêtre, Paris, France. [4] Metabolomics and Cell Biology Platforms, Gustave Roussy Comprehensive Cancer Institute, Villejuif, France. [5] Sorbonne Université, Inserm, CNRS, Centre d'Immunologie et des Maladies Infectieuses CIMI, Paris, France. [6] Sotio, Prague, Czech Republic. [7] Department of Immunology, Charles University, 2nd Faculty of Medicine and University Hospital Motol, Prague, Czech Republic. [8] Caryl and Israel Englander Institute for Precision Medicine, New York, NY, USA. [9] Institute of Medical Immunology, Martin Luther University Halle-Wittenberg, Halle, Germany. [10] Sandra and Edward Meyer Cancer Center, New York, NY, USA. [11] Gustave Roussy Cancer Center, Villejuif, France. [12] INSERM U1015, Villejuif, France. [13] Center of Clinical Investigations in Biotherapies of Cancer (CICBT) 1428, Villejuif, France. [14] Department of Cancer Medicine, Gustave Roussy Cancer Center, Villejuif, France. [15] Pôle de Biologie, Hôpital Européen Georges Pompidou, AP-HP, Paris, France. [16] Suzhou Institute for Systems Medicine, Chinese Academy of Medical Sciences, Suzhou, China. [17] Department of Women's and Children's Health, Karolinska University Hospital, Stockholm, Sweden. [18] Department of Dermatology, Yale School of Medicine, New Haven, CT, USA. [19] Université de Paris, Paris, France. [20] These authors jointly supervised: Guido Kroemer, Lorenzo Galluzzi. ✉email: kroemer@orange.fr; deadoc80@gmail.com

Hormone receptor (HR)$^+$ breast cancer (BC) causes the majority of BC-related deaths in the US and Europe[1]. Most newly diagnosed patients with HR$^+$ BC patients are managed by initial surgery, followed by adjuvant endocrine therapy ± radiotherapy. Although gene expression profiles are used to predict benefit from adjuvant anthracycline-, cyclophosphamide- and taxane-based chemotherapy[2], risk assessment remains imprecise, and treatment efficacy is variable, resulting in many women being treated to benefit a small number. The recognition that the individual benefits of chemotherapy are limited and that current therapies are associated with considerable side effects[3] has shifted BC research toward novel approaches, including immunotherapy.

Immune checkpoint blockers (ICBs) have been successfully implemented in the management of multiple tumors[4,5]. Accumulating evidence indicates that the immune system also plays a major role in the control of mammary carcinogenesis and tumor progression[6]. Accordingly, ICBs targeting programmed cell death 1 (PDCD1; best known as PD-1) or CD274 (the main PD-1 ligand, best known as PD-L1) have recently been suggested to constitute good therapeutic options for triple-negative BC (TNBC) patients[7]. However, the clinical experience with ICBs in HR$^+$ BC patients has been disappointing[8]. At least in part, this reflects the lack of adequate preclinical models that recapitulate the incidence, natural progression, and response to therapy of HR$^+$ BC in immunologically competent hosts. Indeed, human xenografts are intrinsically inadequate for studying tumor-targeting immunity in mice, and mouse HR$^+$ BC cell lines are implanted after being immunoedited by their original host (and hence have already evaded immune recognition)[9]. Similarly, transgenic BC models often bear rather few non-synonymous mutations, and hence are largely invisible to the adaptive immune system[9].

In an attempt to circumvent these issues, we focused on endogenous BC driven in mice by slow-release medroxyprogesterone acetate (MPA, M) pellets combined with an oral carcinogen, 7,12-dimethylbenz[a]anthracene (DMBA, D)[10]. MPA is a multipurpose synthetic progestin that has been associated with increased incidence of BC when used as part of hormone replacement therapy in post-menopausal women[11]. DMBA is a polycyclic aromatic hydrocarbon (PAH) found in cigarette smoke, industrial pollution and grilled meat, and exposure to these risk factors correlates with increased BC incidence[12].

Here, we report the results of in-depth biological, immunological, and functional studies demonstrating that M/D-driven mammary tumors recapitulate several biological and immunological features of human luminal B (highly proliferative HR$^+$HER2$^-$) BC, hence constituting a privileged platform to investigate therapeutic strategies with translational potential. Harnessing this model, we demonstrate that nicotinamide (NAM)—a variant of vitamin B$_3$ currently sold over the counter as nutritional supplement—not only delays the development of endogenous mammary carcinogenesis, but also exerts immunotherapeutic effects against M/D-driven tumors and other transplantable mouse models of luminal B BC. The oncopreventive activity of NAM overcomes the detrimental effect of other common risk factors for BC, including a high-fat diet (HFD), and relies on the reinstatement of T cell-dependent immunosurveillance, which is otherwise not operational during oncogenesis. These findings identify NAM supplementation as a potential strategy for the prevention and treatment of HR$^+$ BC.

## Results

**M/D-driven tumors recapitulate key features of human HR$^+$ BC.** When 6–9-week-old female C57BL/6 mice were implanted with a slow-release pellet liberating MPA (day 0) and received 6 oral gavages of the environmental pollutant DMBA (administered 1, 2, 3, 5, 6, and 7 weeks after implantation of the MPA pellet), they developed mammary tumors with a highly variable latency (day ~50–150) with a 50% tumor incidence around day 90–110 (Fig. 1a, b). Similar findings were recorded for BALB/c female mice (Supplementary Fig. 1a). M/D-driven mammary tumors occurred at a slightly higher incidence at inguinoabdominal vs cervicothoracic sites (Supplementary Fig. 1b), initially as single lesions but later as multifocal disease (at least in a fraction of cases, Supplementary Fig. 1c), and progressed rapidly if untreated (Fig. 1b).

M/D-driven tumors often displayed: (1) a relatively monotonous population of cells, poorly circumscribed, infiltrating the surround soft and adipose tissues in a loose pattern, (2) cords and nodules of atypical epithelial cells, with some duct or gland formation, and (3) some degree of vascular and lymphatic permeation (Fig. 1c). Thus, M/D-driven tumors histologically resemble to human HR$^+$ BCs. Transcriptomic studies based on The Cancer Genome Atlas (TCGA) database for human BC as a reference revealed that M/D-driven tumors display a striking similarity to highly proliferative HR$^+$HER2$^-$ (luminal B) human BC (Fig. 1d and Supplementary Fig. 1d). Immunofluorescence microscopy confirmed that untreated M/D-driven tumors express estrogen receptor 1 (ESR1, best known as ER) but not vimentin (VIM, a marker of basal BC) (Supplementary Fig. 1e). Accordingly, M/D-driven tumors emerged with delayed kinetics in mice bearing a deletion in Esr1 causing defective transcriptional activity (ER-AF2$^0$)[13] (Fig. 1e), while oncogenesis and tumor progression were accelerated in mice expressing an ER mutant associated with increased nuclear accumulation (ER$^{C451A}$)[14] (Fig. 1f). The ER antagonist tamoxifen delayed M/D-driven oncogenesis and cancer-related death, at least during the early stages of the process (Fig. 1g). Comparing the transcriptome of normal mouse mammary glands, hyperplastic glands exposed to M/D but not developing tumors, and M/D-driven tumors revealed major changes linked to malignant transformation, including the upregulation of proliferation-related genes as well as in the downregulation of transcripts linked to immune functions (Fig. 1h). In particular, transcripts associated with germinal centers (Btla, Cd4, Cd19, Cd22, Dock10, H2-M2) and tertiary lymphoid organs (Ccl21a, Ccl21b, Cd3d, Cd3e, Cd3g, Cxcl13, Dpp4, Il7r, Sell, Tcrb) were markedly downregulated during oncogenesis (Fig. 1h). Consistently, established M/D-driven tumors exhibited limited infiltration by CD3$^+$CD8$^+$ cytotoxic T lymphocytes (CTLs) and CD3$^+$CD4$^+$ T cells, as compared to normal mammary glands (Fig. 1i), and their CD4$^+$ compartment was enriched in immunosuppressive FOXP3$^+$ regulatory T (T$_{REG}$) cells (Fig. 1i). Thus, M/D-driven tumors established in immunocompetent mice recapitulate key aspects of human luminal B BC, which is also scarcely infiltrated by immune cells[6].

**M/D-driven tumors are under immunosurveillance by NK cells.** Based on transcriptomic and cytofluorometric data (Fig. 1h, i), we postulated that M/D-driven tumors would evolve by escaping immunosurveillance. Indeed, M/D-driven carcinogenesis was accelerated, and overall survival (OS) was shortened, in Rag2$^{-/-}$ Il2rg$^{-/-}$ mice (lacking T cells, B cells and NK cells) (Fig. 2a), as well as in mice lacking interferon gamma (Ifng, encoding a major effector of lymphoid immunosurveillance) (Fig. 2b). To characterize the effects of immune components on the emergence of HR$^+$ tumors, we performed a large-scale experiment in which we compared tumor-free survival (TFS) and OS following M/D-driven oncogenesis between untreated (or sham-treated) wild-type

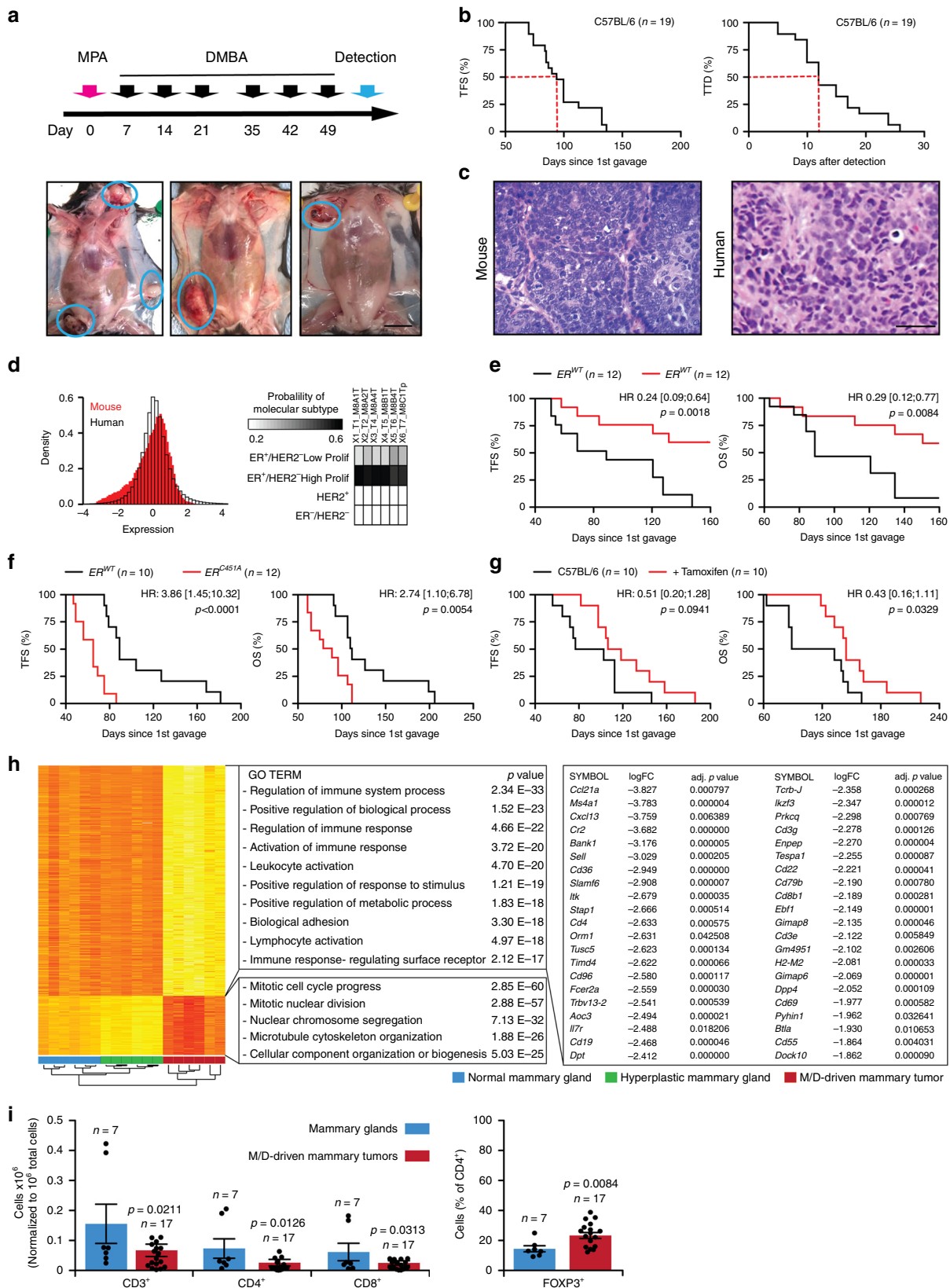

(WT) BALB/c or C57BL/6 mice and: (1) mice bearing a range of different inherited defects in innate and adaptive immune functions (genotypes: $Casp1^{-/-}$, $Il21^{-/-}$, $Il22^{-/-}$, $Il21r^{-/-}$, $Il1r1^{-/-}$, $Myd88^{-/-}$, $Nlrp3^{-/-}$, $Rag2^{-/-}$); (2) WT mice treated with agents that neutralize specific cytokines (i.e., IFNG, IL17) or cytokine receptors (i.e., IL1R, IFNAR1); or (3) WT mice depleted of CD4$^+$ and CD8$^+$ cells, or NK cells. This experiment confirmed that M/D-driven carcinogenesis is under the control of IFNG-producing cells and revealed that the absence of $Il21$ or $Il22$, the blockade of IFNAR1, and the depletion of NKG2D$^+$ cells also precipitate

**Fig. 1 M/D-driven tumors recapitulate key features of human HR+ breast cancer. a** Schedule of oncogenic challenges for the induction of M/D-driven tumors and representative images of M/D-driven tumors established in WT C57BL/6 mice. Scale bar = 1 cm. **b** Tumor-free survival (TFS) and time-to-death (TTD) in WT C57BL/6 mice subjected to M/D-driven oncogenesis. Number of mice is reported. **c** Representative histology of M/D-driven tumors as compared to human HR+ breast cancers. Scale bar = 50 μm. **d** Probeset distribution comparing the transcriptomic profile of 6 M/D-driven mammary tumors established in C57BL/6 mice with that of human breast cancers from the TCGA public database. Probability of molecular subtyping is reported for each mouse tumors. **e, f** TFS and overall survival (OS) of WT C57BL/6 mice and C57BL/6 bearing ER mutations that cause nuclear exclusion (**e**) or nuclear accumulation (**f**) subjected to M/D-driven oncogenesis. Number of mice, hazard ratio (HR) and p values (two-sided log-rank) are reported. **g** TFS and OS of WT C57BL/6 mice subjected to M/D-driven oncogenesis in control conditions or along with tamoxifen administration in the drinking water. Number of mice, HR and p values (two-sided log-rank) are reported. **h.** Non-supervised hierarchical clustering of the transcriptomic profile of M/D-driven tumors established in WT C57BL/6 mice (n = 6), mammary glands exposed to M/D but not developing tumors (n = 6) and M/D-naïve mammary glands (n = 6). Gene Ontology analysis, fold change (FC) and adjusted, two-sided p values are reported. Red, upregulation. Yellow, downregulation. **i** Relative amount of CD3+, CD8+, CD4+ and CD25+FOXP3+ cells infiltrating M/D-driven tumors in C57BL/6 mice vs syngeneic M/D-naïve mammary glands. Results are means ± SEM plus individual data points. Number of mice and p values (unpaired, two-sided Student's t, as compared to M/D-naïve mammary glands) are reported.

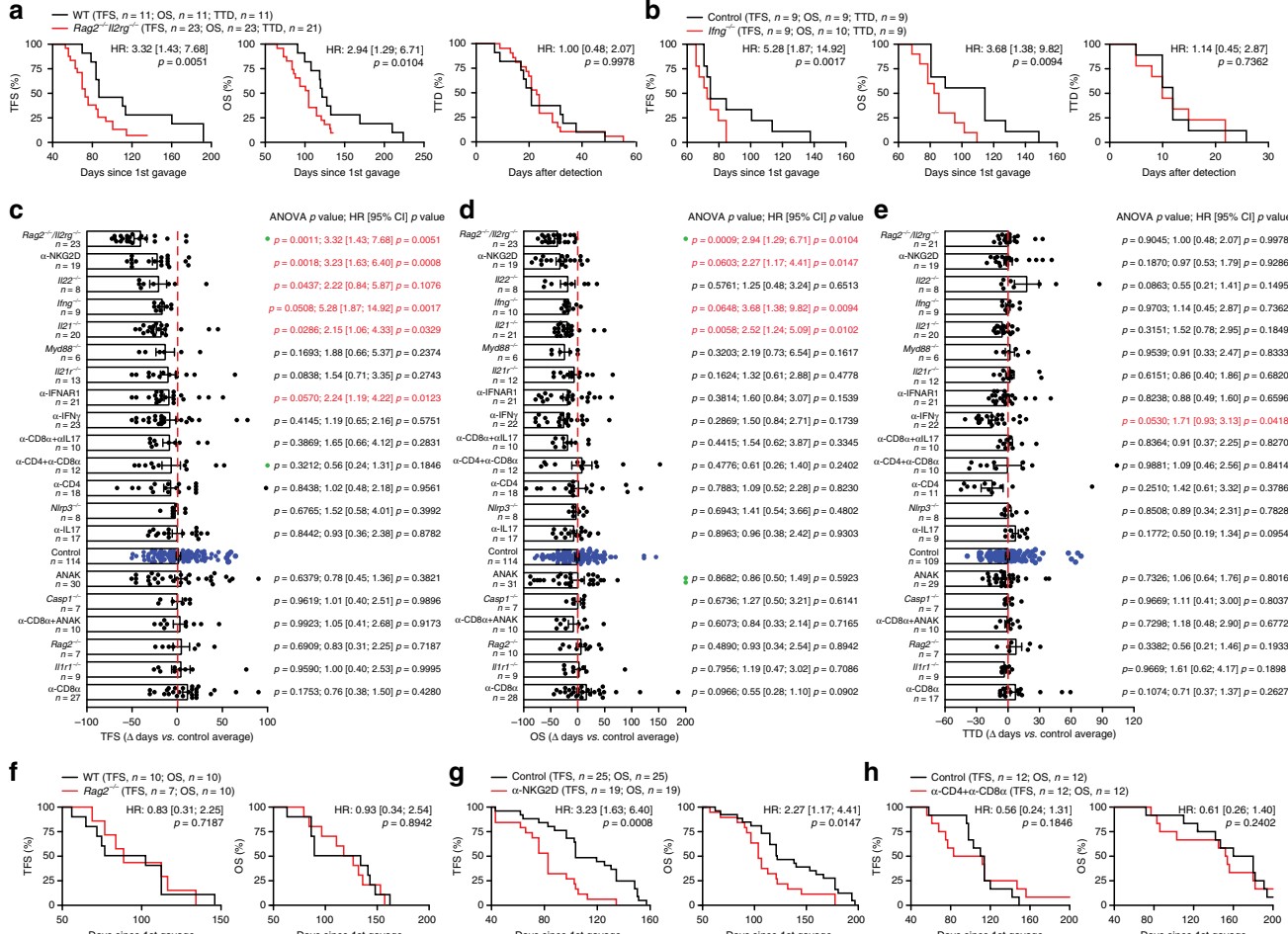

**Fig. 2 M/D-driven oncogenesis is under immunosurveillance by NK cells. a, b** Tumor-free survival (TFS), overall survival (OS) and time-to-death (TTD) of WT C57BL/6 mice (**a, b**) and Rag2−/−Il2rg−/− mice (**a**) or Ifng−/− mice (**b**) subjected to M/D-driven oncogenesis. Number of mice, hazard ratio (HR) and p values (two-sided log-rank) are reported. **c–e.** Variations in TFS (**c**), OS (**d**) and TTD (**e**) imposed to M/D-driven oncogenesis in mice by the indicated genotype or immunomodulatory interventions. Results are means ± SEM plus individual data points. Number of mice, HR and p values (two-sided log-rank and one way-ANOVA plus Fisher LSD, calculated with respect to individual control experiments) are reported. Green dots indicate mice that were free of disease (**c**) and alive (**d**) at the end of the experiment. **f–h** TFS and OS of WT C57BL/6 (**f–h**) mice and Rag2−/− BALB/c mice (**f**), C57BL/6 mice receiving NKG2D-depleting antibodies (**g**), or C57BL/6 mice receiving CD4- and CD8-depleting antibodies (**h**). Number of mice, HR and p values (two-sided log-rank) are reported.

oncogenesis in this setting (Fig. 2c). These effects were also manifest at the level of OS for the Ifng−/− and Il21−/− genotypes and for NKG2D+ cell depletion (Fig. 2d). Disease progression computed as time from detection to death (time to death, TTD) was only accelerated by IFNG-neutralizing antibodies (Fig. 2e).

Importantly, while M/D-driven oncogenesis was fostered by a severe immunodeficiency affecting T lymphocytes, B cells and NK cells (Rag2−/−Il2rg−/− genotype), TFS and OS in Rag2−/− mice (who lack T lymphocytes and B cells, but have an intact NK cell compartment)[9] did not differ from TFS and OS in WT mice

(Fig. 2f). Moreover, TFS and OS were negatively affected by the depletion of NKG2D$^+$ cells (encompassing NK cells and T cells), but not by the co-depletion of CD4$^+$ and CD8$^+$ lymphocytes (Fig. 2g, h). Altogether, these findings establish the importance of immunoevasion for the emergence and progression of HR$^+$ BC, placing emphasis on interferon signaling in the context of NK cell-mediated immunosurveillance. In further support of this notion, the partial oncopreventive effect of tamoxifen was lost in $Rag2^{-/-}Il2rg^{-/-}$ mice (Supplementary Fig. 1f).

**M/D-driven tumors are poorly immunoedited by T cells.** To further explore immunosurveillance in HR$^+$ BC, we established cell lines from M/D-driven tumors developing in immunocompetent mice. When injected subcutaneously (s.c.) or into the mammary fat pad of immunodeficient $Rag2^{-/-}$ mice, these cell lines rapidly generated cancers with aggressive, MPA-insensitive proliferation (Fig. 3a, b). In sharp contrast, these cell lines only sporadically (in 2 out of 52 attempts) generated progressive lesions upon s.c. injection into immunocompetent, syngeneic C57BL/6 hosts (Fig. 3c), and invariably failed to establish progressing neoplasms upon orthotopic implantation (Fig. 3d), irrespective of whether mice received MPA. Often, small tumors (generally <50 mm$^2$ surface) formed, but spontaneously regressed over time (Fig. 3a–d). DMBA administration, alone or in combination with MPA, failed to promote the growth of M/D-driven cancer cells injected s.c. in syngeneic immunocompetent recipients, although it caused the development of endogenous tumors (in 5/20 mice with DMBA alone, and 20/20 mice with DMBA + MPA) (Fig. 3e), implying that DMBA does not cause severe immunodeficiency in this model. As an additional control, mammary cancer cell lines established from M/D-driven tumors evolving in $Rag2^{-/-}$ mice always failed to develop tumors when transplanted into WT mice (Fig. 3f).

**Prophylactic vaccination delays M/D-driven carcinogenesis.** We next investigated whether vaccinating immunocompetent mice with cell lines established from M/D-driven cancers would have a preventive effect on M/D-driven oncogenesis. To this aim, cell lines established from M/D-driven tumors evolving in $Rag2^{-/-}$ or WT mice were killed in vitro with the immunogenic cell death inducer mitoxantrone (MTX)[15,16] and injected into the fat pad of WT mice 1 week prior to, in the 4th week of, and 2 days after de novo M/D-driven carcinogenesis. This treatment postponed the manifestation of M/D-driven tumors irrespective of the immunological status of the host ($Rag2^{-/-}$ vs WT) from which cell lines were derived (Fig. 3g). Importantly, injection of organoids from the normal breast epithelium—optionally treated with MTX or radiation therapy (another inducer of immunogenic cell death)[15,16]—had a similar capacity to delay M/D-driven oncogenesis in C57BL/6 mice (Fig. 3h, i). In this setting, neutralization of annexin A1 (ANXA1)—a danger signal involved in the perception of cell death as immunogenic[15,16]—abolished the vaccination effect (Fig. 3i). Thus, immunosurveillance of M/D-driven carcinogenesis can be boosted, at least to some degree, by immunological interventions including vaccination.

**Safe nutritional measures delay M/D-driven carcinogenesis.** Multiple epidemiological studies indicate that nutritional status has a major impact on BC incidence[17], and it is now established that dietary interventions can mediate therapeutically relevant immunostimulatory effects[18]. We therefore analyzed the impact of nutritional interventions on M/D-driven carcinogenesis in WT C57BL/6 mice. We found that biweekly 24-h-long fasting cycles postponed M/D-driven oncogenesis, decelerated disease progression and hence extended OS (Fig. 4a), although they did not

cause a stable reduction in body mass as compared to unrestricted access to standard chow (Supplementary Fig. 1g). Along similar lines, dietary supplementation with NAM (Fig. 4b) and nicotinamide riboside (NR), which are two variants of vitamin B$_3$, pyridoxine (vitamin B$_6$) and calcitriol (vitamin D$_3$) all tended to ameliorate TFS and OS in WT C57BL/6 subjected to M/D-driven carcinogenesis (Fig. 4c–e). In contrast, dietary supplementation with spermidine (a polyamine with prominent life-extending effects)[19] failed to affect M/D-driven oncogenesis (Fig. 4c–e). Amongst all vitamins tested, NAM delivered with the drinking water (which was more active than the administration of NR with chow) was the only one to completely (>300 days) block oncogenesis in a considerable fraction (~25%) of mice, and to robustly delay disease progression (Fig. 4a–e). Of note, dietary supplementation of NAM to C57BL/6 mice resulted in circulating NAM levels of ~16.3 µg mL$^{-1}$ µM (Supplementary Fig. 1h), while NAM concentrations around 2.3 µg mL$^{-1}$ have been documented in healthy volunteers receiving 2 g nicotinic acid (a precursor of NAM) per os[20]. The beneficial effects of NAM could also be documented in a transgenic model of immunoevasive, luminal B BC driven by the polyomavirus middle T (PyMT) antigen expressed under control of the MMTV promoter[21,22] (Supplementary Fig. 1i).

A high sucrose diet (HSD) accelerated M/D-driven oncogenesis, an effect that was completely abolished by concomitant NAM supplementation, which continued to extend TFS and OS beyond the level of control mice fed a normal chow (although only one mice remained cancer-free) (Fig. 4c–e). Less prominent beneficial effects on TFS, OS or TTD were observed when HSD was provided in the context of pyridoxine or calcitriol supplementation (Fig. 4c–e). A dietary regimen rich in fat (high-fat diet, HFD) accelerated M/D-driven carcinogenesis and shortened OS, but had no impact on TTD (Fig. 4c–e). Such a detrimental effect of HFD was fully antagonized by NAM but not by calcitriol or pyridoxine (Fig. 4c–e). In this context, NAM also reversed the increase in body weight caused by HFD (Supplementary Fig. 1f). In conclusion, M/D-driven tumorigenesis is sensitive to dietary alterations with an established detrimental effect on human BC, including HSD and HFD, and NAM mediates robust oncopreventive effects in all these scenarios.

**NAM mediates chemopreventive effects that depend on T cells.** We observed that the oncopreventive effects of NAM, pyridoxine and calcitriol were invariably lost in $Rag2^{-/-}Il2rg^{-/-}$ mice (Fig. 5a–c). Moreover, NAM-dependent oncoprevention was abolished in $Rag2^{-/-}$ mice, in mice subjected to the antibody-mediated co-depletion of CD4$^+$ and CD8$^+$ T cells, in mice receiving an IFNG-neutralizing antibody, as well as in $Ifnar1^{-/-}$ mice (Fig. 5d–g). These data lend further support to the notion that interferon signaling is key for the immunosurveillance of HR$^+$ BC. Moreover, they demonstrate that nutritional interventions with oncopreventive effects in this model, such as NAM supplementation, boost T cell-dependent immunosurveillance.

NAM has been suggested to trigger autophagy, a cytoprotective pathway involving the lysosomal degradation of disposable or potentially dangerous cytosolic entities[23]. However, we failed to detect consistent biochemical signs of autophagy, such as the lipidation of microtubule associated protein 1 light chain 3 beta (MAP1LC3B, best known as LC3) and the degradation of sequestosome 1 (SQSTM1, best known as p62) in M/D-driven tumors developing despite NAM administration, as well as in multiple healthy tissues (Supplementary Fig. 2a). Moreover, the oncopreventive effects of NAM were conserved in $Becn1^{+/-}$ mice (which exhibit partial autophagic defects and increased susceptibility to mammary carcinogenesis)[24,25] (Supplementary Fig. 2b).

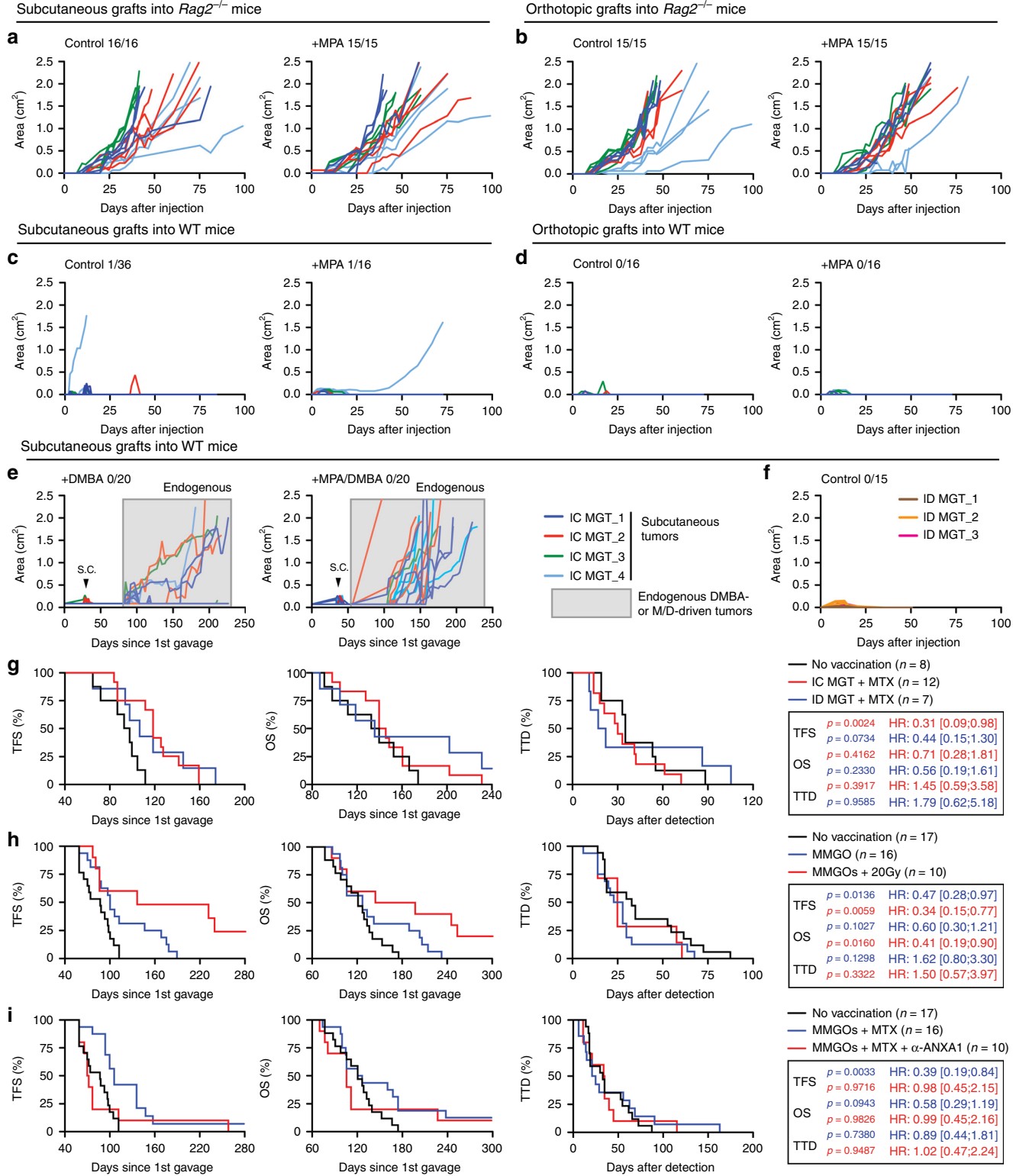

Similarly, M/D-driven oncogenesis remained sensitive to NAM-mediated oncoprevention in mice subjected to the conditional knockout of *Atg5* (a key component of the autophagy machinery)[26] in keratin 5 (KRT5)-expressing cells (including cells of the mammary epithelium)[27] (Supplementary Fig. 2c), indicating that this effect does not require autophagy induction in malignant cells. To further explore the immunostimulatory activity of NAM, tumor-free C57BL/6 mice were treated with NAM for 1, 3 or 14 days, followed by RNAseq analyses of splenocytes and lymph

node cells. In this setting, we detected early upregulation of gene sets involved in immune functions, notably type I interferon signaling and chemotaxis (Fig. 5h, i). However, we were unable to detect consistent quantitative or proliferative alterations in the CD8$^+$ T cell, CD4$^+$ T cell, CD4$^+$FOXP3$^+$ regulatory T (T$_{REG}$), NK cell, NKT cell, and B cell compartments of the spleen and lymph nodes of tumor-naïve C57BL/6 mice receiving NAM in the drinking water for 14 days (Supplementary Fig. 2d, e), suggesting that the ability of NAM to delay M/D-driven carcinogenesis

**Fig. 3 M/D-driven tumors are poorly immunoedited by T cells and sensitive to vaccination. a–d** Growth of cell lines established from M/D-driven tumors (herein called mammary gland tumors, MGTs) developing in immunocompetent (IC) mice upon subcutaneous (**a**, **c**) or orthotopic (**b**, **d**) implantation in *Rag2⁻/⁻* (**a**, **b**) or IC C57BL/6 mice (**c**, **d**) optionally bearing an MPA-releasing pellet. Individual curves and tumor incidence are reported. **e** Growth of MGTs upon subcutaneous implantation in IC C57BL/6 mice receiving DMBA alone or in the context of an MPA-releasing pellet. Individual curves for all tumors and subcutaneous tumor incidence are reported. Boxed curves refer to endogenous tumors caused by DMBA alone or MPA plus DMBA. **f** Growth of MGTs established from M/D-driven tumors developing in immunodeficient (ID) *Rag2⁻/⁻* mice upon subcutaneous implantation in IC C57BL/6 mice. Individual curves and global tumor incidence are reported. **g** Tumor-free survival (TFS), overall survival (OS) and time-to-death (TTD) of WT C57BL/6 mice subjected to M/D-driven oncogenesis in control conditions or upon vaccination with cell lines established from M/D-driven tumors developing in IC or ID mice and treated with mitoxantrone (MTX) in vitro. Number of mice, hazard ratio (HR) and p values (two-sided log-rank, as compared to non-vaccinated mice) are reported. **h** TFS, OS, and TTD of WT C57BL/6 mice subjected to M/D-driven oncogenesis in control conditions or upon vaccination with mouse mammary gland organoids (MMGOs) optionally killed in vitro by exposure to radiation therapy (RT) in a single dose of 20 Gy. Number of mice, HR and p values (two-sided log-rank, as compared to non-vaccinated mice) are reported. Please note that part of these results (control conditions) are also depicted in (**i**). **i** TFS, OS, and TTD of WT C57BL/6 mice subjected to M/D-driven oncogenesis in control conditions or upon vaccination with MMGOs killed in vitro by mitoxantrone (MTX) administration in the optional context of ANXA1 neutralization. Number of mice, HR and p values (two-sided log-rank, as compared to non-vaccinated mice) are reported. Please note that part of these results (control conditions) are also depicted in (**h**).

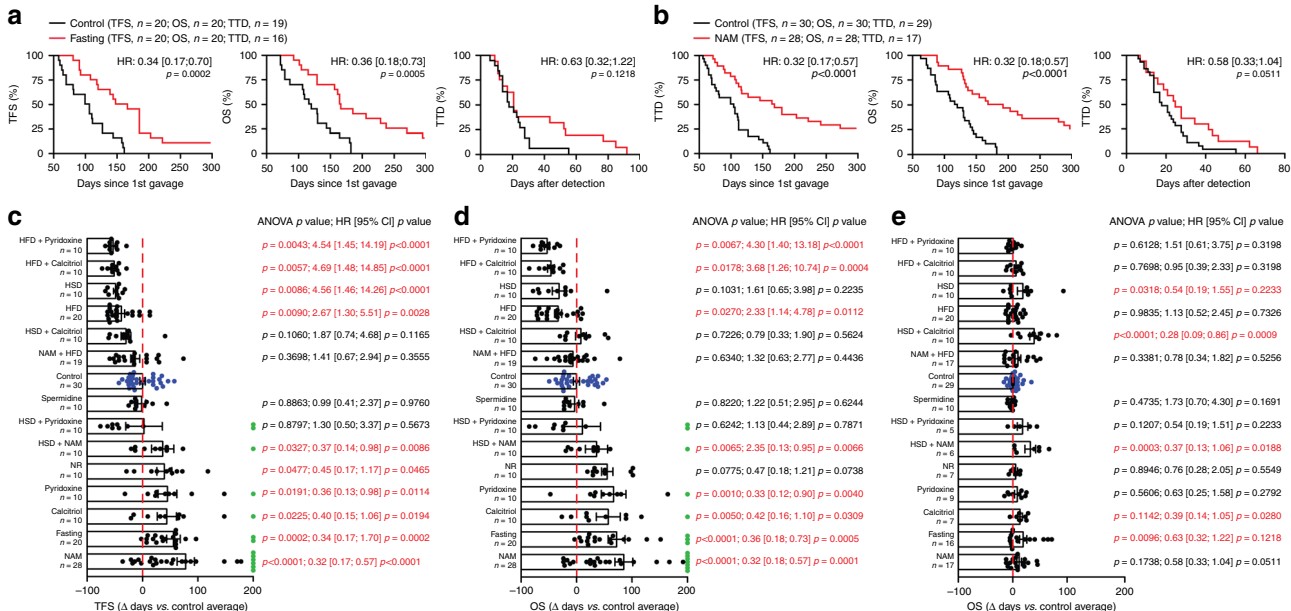

**Fig. 4 Safe nutritional interventions delay M/D-driven oncogenesis. a**, **b** Tumor-free survival (TFS), overall survival (OS) and time-to-death (TTD) of C57BL/6 mice subjected to M/D-driven oncogenesis in control conditions (**a**, **b**) or in the context of weekly 24-h-fasting episodes (**a**) or NAM supplementation with the drinking water (**b**). Number of mice, hazard ratio (HR) and p values (two-sided log-rank) are reported. **c–e** Variations in TFS (**c**), OS (**d**) and TTD (**e**) imposed to M/D-driven oncogenesis in C57BL/6 mice by the indicated nutritional interventions. Results are means ± SEM plus individual data points. Number of mice, HR and p values (two-sided log-rank and one way-ANOVA plus Fisher LSD, calculated with respect to individual control experiments) are reported. Green dots indicate mice that were free of disease (**c**) and alive (**d**) at the end of the experiment. HFD high-fat diet, HSD high sucrose diet, NR nicotinamide riboside.

involves immunological pathways not linked to the systemic expansion/contraction of immune effectors. In summary, NAM appears to mediate oncopreventive effects through an autophagy-independent, immunological mechanism that relies on lymphoid cells (T cells and NK cells) as well as interferon signaling.

**NAM mediates immunotherapeutic activity against BC.** Continuous NAM supplementation delayed oncogenesis and hence prolonged OS in mice exposed to M/D-driven carcinogenesis (Fig. 4b–e). When dietary NAM supplementation was started at (rather than before) tumor detection, NAM-mediated therapeutic effects that were more pronounced than those observed with continuous NAM provision from oncogenesis (Fig. 6a, b), and were not associated with the loss of the luminal B (ER⁺VIM⁻) phenotype (Supplementary Fig. 3a). NAM-mediated therapeutic activity also in transplantable models of luminal B BC (AT3 and

TSA cells)²⁸,²⁹ as well as in transplantable fibrosarcoma sarcoma models (MCA205 cells) established in immunocompetent syngeneic hosts (Supplementary Fig. 3b–d). This effect did not depend on autophagy activation, as demonstrated in AT3 cells subjected to the shRNA-dependent depletion of ATG7 (a core component of the autophagy machinery)³⁰, as well as in *Atg7⁻/⁻* TSA cells (Supplementary Fig. 3e, f).

To mechanistically explore the therapeutic effects of NAM, we compared the transcriptomic profile of untreated vs NAM-treated M/D-driven tumors, revealing signs of immune regulation (Fig. 6c). In particular, NAM administration restored some of the physiological transcriptional features that were lost in the course of M/D-driven oncogenesis (Fig. 1h), such as the expression of genes associated with germinal centers (*Cd22*, *Dock10*, *Gimap8*) (Fig. 6c), and—to a lesser degree—lymphoid cell activation (*Cd48*, fold change = 2.23, p = 1.62E−02; *Cd160*,

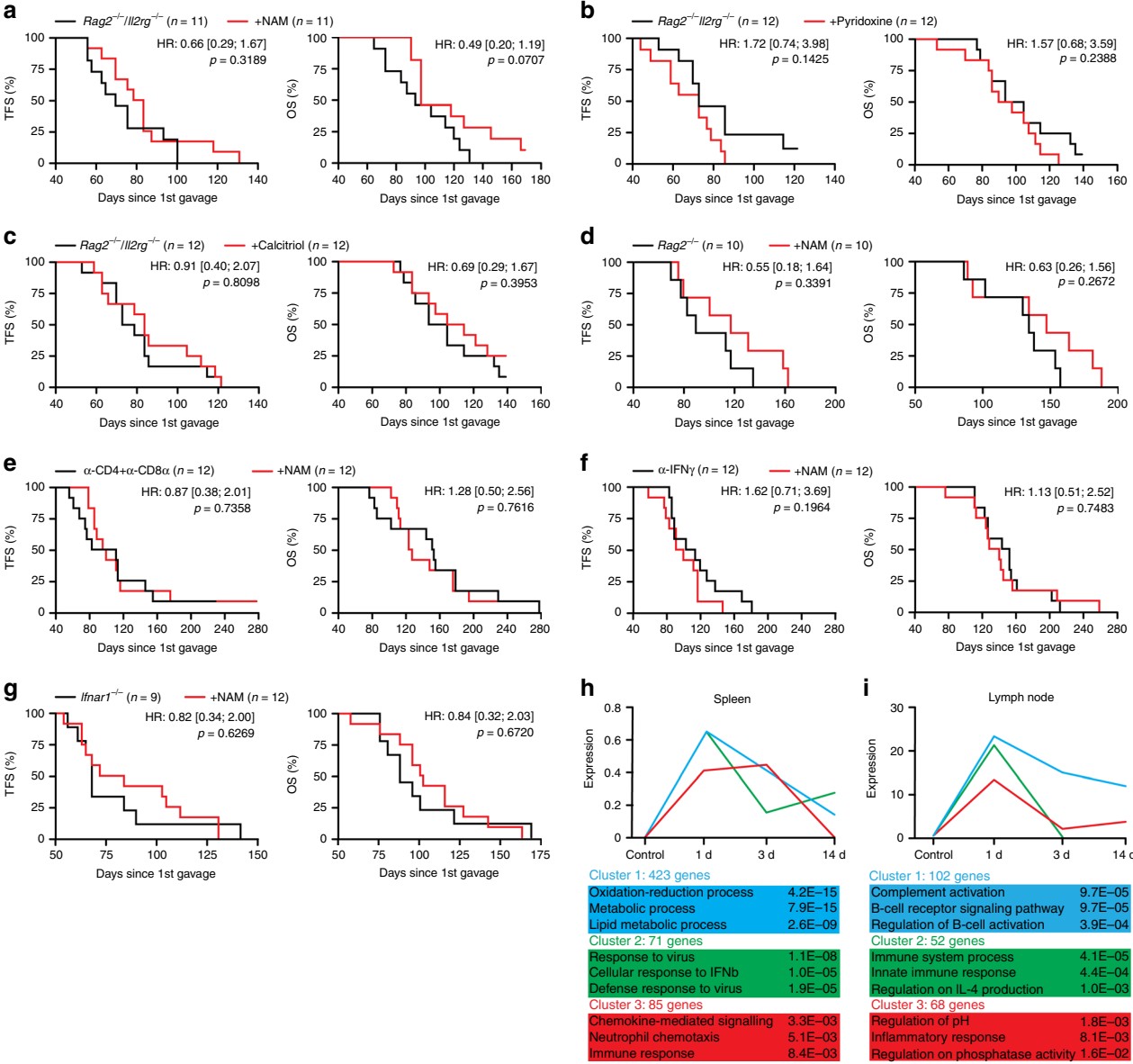

**Fig. 5 NAM mediates chemopreventive effects by boosting immunosurveillance. a–c** Tumor-free survival (TFS) and overall survival (OS) of *Rag2−/−Il2rg−/−* mice subjected to M/D-driven oncogenesis in control conditions (**a–c**) or along with NAM (**a**), pyridoxine (**b**) or calcitriol (**c**) supplementation. Number of mice, hazard ratio (HR) and *p* values (two-sided log-rank) are reported. **d–g** TFS and OS of *Rag2−/−* mice (**d**), C57BL/6 mice receiving CD4- and CD8α-depleting antibodies (**e**), C57BL/6 mice receiving an IFNG-neutralizing antibody (**f**), and *Ifnar1−/−* mice (**g**) subjected to M/D-driven oncogenesis in control conditions or along with NAM supplementation with the drinking water. Number of mice, HR and *p* values (two-sided log-rank) are reported. **h, i** Gene sets enriched in the spleen (**h**) and lymph nodes (**i**) of tumor-naive C57BL/6 mice receiving NAM supplementation with the drinking water for 1, 3 or 14 days. Adjusted, two-sided *p* values for enrichment are reported.

fold change = 2.37, *p* = 4.23E−02; *Cd244,* fold change = 2.84, *p* = 5.07E−03). A similar study of untreated *vs* NAM-treated AT3 tumors identified transcriptional patterns linked to T-cell differentiation, as well as to type I and type III interferon signaling (Supplementary Fig. 3g). Only 23 genes were consistently upregulated in both M/D-driven and AT3 mammary tumors responding to NAM (Fig. 6d and Supplementary Fig. 3g). Ingenuity pathway analysis indicated that most of these genes positively regulate immunological processes including the recognition of antigenic determinants (TCR signaling, BCR signaling, co-receptor signaling), as well as NK cell, interleukin and chemokine signaling (Fig. 6e). However, when we compared the immunological infiltrate of untreated M/D-driven tumors at 0.7–1 cm² surface area with that of their NAM-treated

counterparts (at a similar size), we failed to identify consistent changes in the abundance of CD8+, CD4+ T cells and T_REG cells, NK and NKT cells, and B cells (Supplementary Fig. 4a). Moreover, we failed to detect significant differences on CD8+, CD4+ T cells and T_REG cells from untreated *vs* NAM-treated tumors with respect to proliferation (Ki67 positivity) (Supplementary Fig. 4a). Similar observations, which we attribute to the considerable heterogeneity of the model, were made in the spleen and lymph nodes of C57BL/6 mice bearing M/D-driven tumors (Supplementary Fig. 4b, c). Conversely, AT3 tumors treated with NAM exhibited increased levels of CD8+ and CD4+ T cells, as well as a higher CD8+/CD4+ T cell ratio as compared to untreated tumors (Fig. 6f). Moreover, NAM-treated AT3 tumors contained increased amounts of CD8+ cells (co)-expressing the

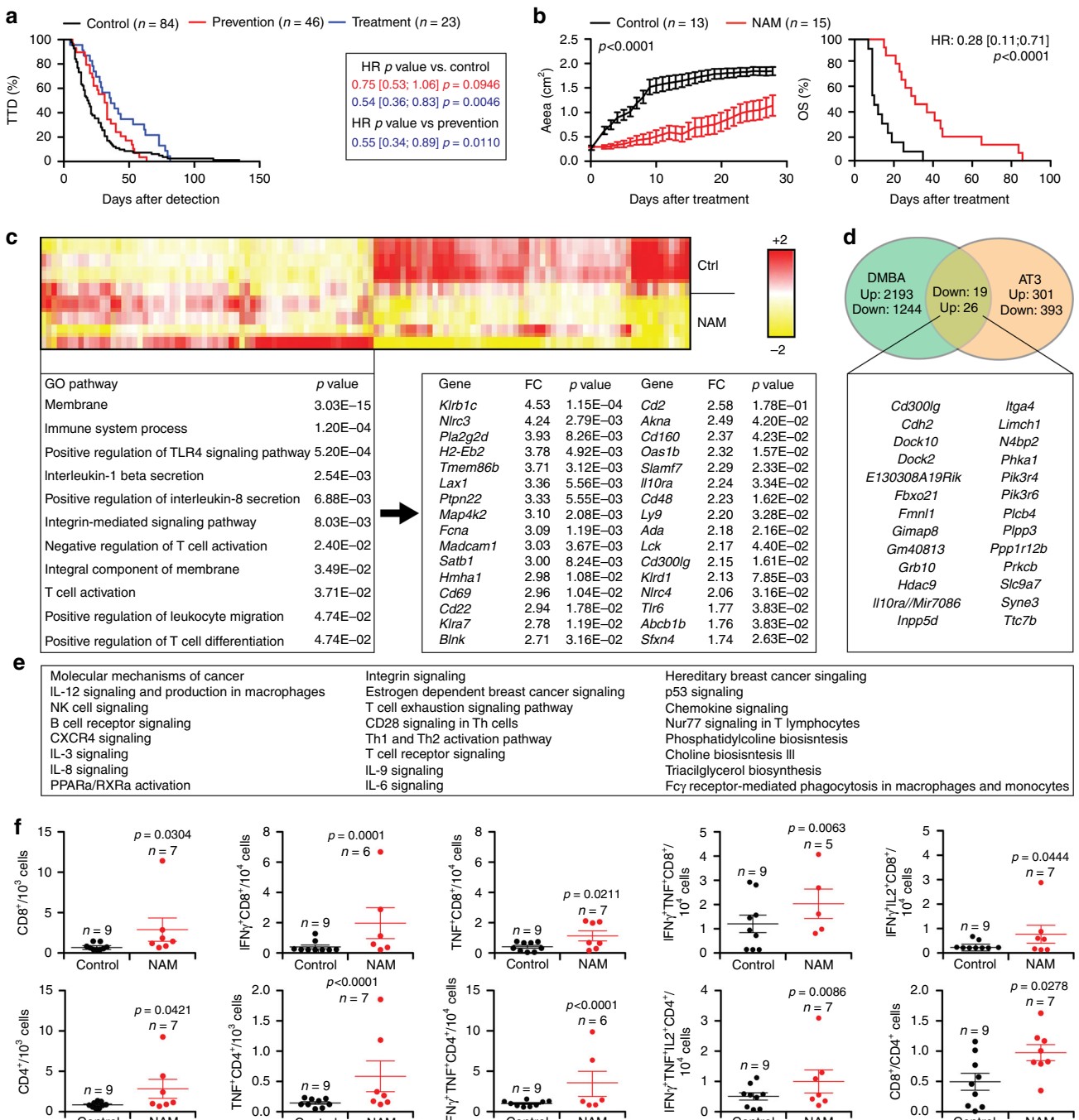

**Fig. 6 NAM mediates immunotherapeutic effects on established BCs. a** Time-to-death (TTD) of C57BL/6 mice subjected to M/D-driven carcinogenesis in control conditions or in the context of NAM supplementation with the drinking water, either from MPA pellet implantation (prevention) or tumor detection (treatment). Number of mice, hazard ratio (HR) and p values (two-sided log-rank) are reported. **b** Tumor growth and overall survival (OS) in C57BL/6 mice bearing established M/D-driven tumors that were maintained in control conditions or subjected to NAM supplementation with the drinking water. Tumor growth results are means ± SEM. Number of mice, HR and p values (two-way ANOVA corrected for row (time) and column (treatment) factors for tumor growth and two-sided log-rank for OS) are reported. **c** Non-supervised hierarchical clustering of genes differentially expressed in untreated (n = 4) vs NAM-treated (n = 4) M/D-driven tumors. Top upregulated genes, fold change (FC) and adjusted, two-sided p values are reported. Gene Ontology analysis and adjusted, two-sided p values for enrichment are indicated. **d** Comparison of private vs shared transcriptional changes induced by NAM treatment in M/D-driven tumors vs AT3 tumors established in C57BL/6 mice. The list of genes upregulated by NAM in both models is provided. See also Supplementary Fig. 3g. **e** Ingenuity Pathway Analysis of genes upregulated by NAM in both M/D-driven tumors AT3 tumors established in C57BL/6 mice. **f** Immune infiltration of AT3 tumors established in C57BL/6 mice that were maintained in control conditions or received NAM supplementation with the drinking water (starting at tumor detection) for 10 days. Results are means ± SEM plus individual data points. Number of mice and p values (one way-ANOVA plus Fisher LSD) are reported.

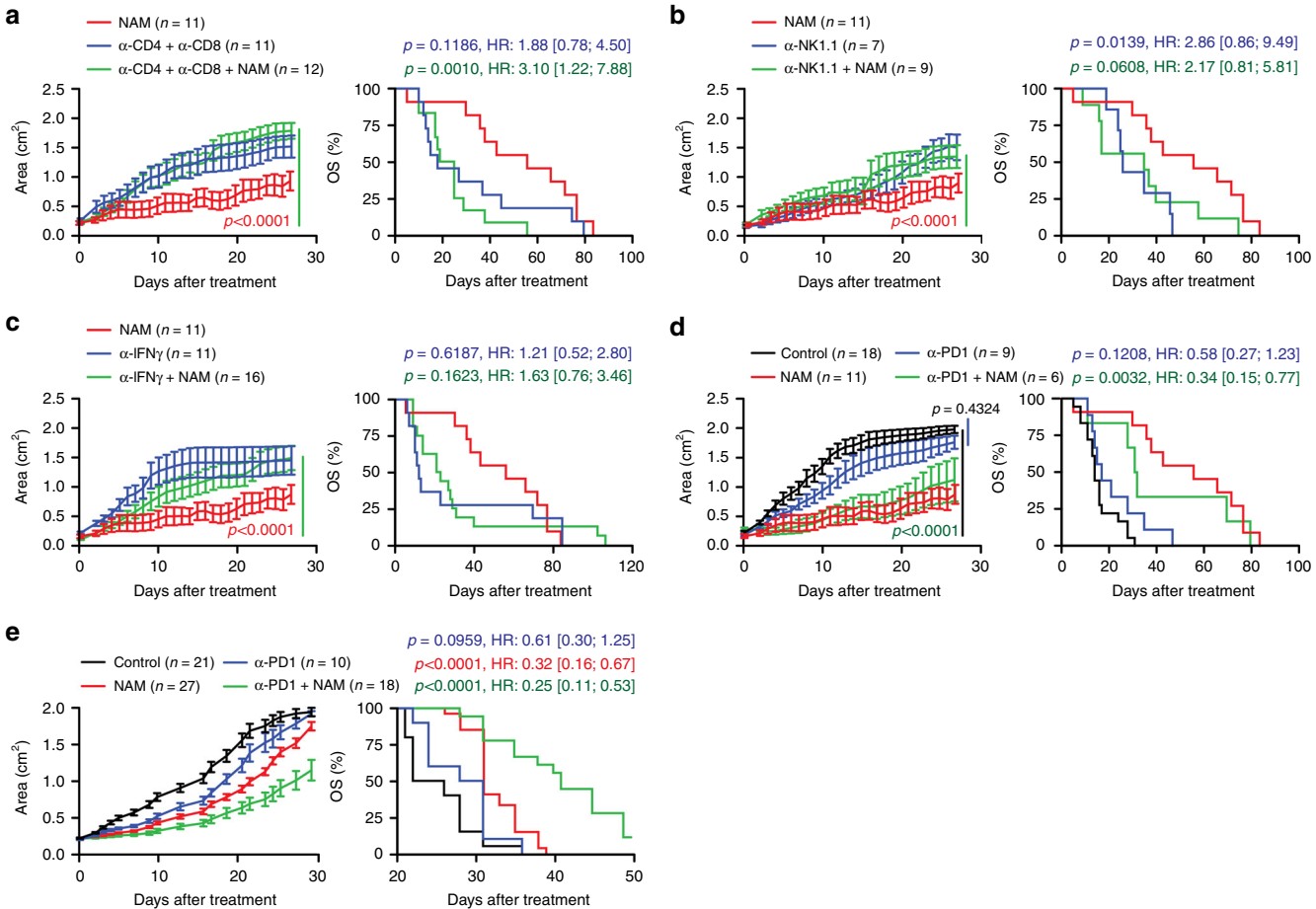

**Fig. 7 The therapeutic effects of NAM depend on T cells and immune signaling. a–c** Tumor growth and overall survival (OS) in C57BL/6 mice bearing established M/D-driven tumors that were maintained in control conditions (**a–c**) or subjected to NAM supplementation with the drinking water alone (**a–c**), or in the context of CD4$^+$ and CD8$^+$ T cell co-depletion (**a**), NK cell depletion (**b**), or IFNG neutralization (**c**). Tumor growth results are means ± SEM. Number of mice, hazard ratio (HR) and p values (two-way ANOVA corrected for row (time) and column (treatment) factors for tumor growth and two-sided log-rank for OS). Please note that NAM-treated tumors (red curves) in (**a–c**) are from the same experiment, also depicted in (**d**). **d, e** Tumor growth and OS in C57BL/6 (**d**) or BALB/c (**e**) mice bearing established M/D-driven (**d**) or TSA (**e**) tumors that were maintained in control conditions or subjected to NAM supplementation with the drinking water or systemic PD-1 blockage, alone or in combination. Tumor growth results are means ± SEM. Number of mice, hazard ratio (HR) and p values (two-way ANOVA corrected for row (time) and column (treatment) factors for tumor growth and two-sided log-rank for OS). Please note that NAM-treated tumors (red curves) in (**d**) are from the same experiment depicted in (**a–c**).

effector molecules IFNG and tumor necrosis factor (TNF), and co-expressing IFNG plus the T cell mitogen interleukin 2 (IL2), as compared to their untreated counterparts (Fig. 6f). Similarly, in the AT3 model, NAM treatment was associated with increased tumor infiltration by CD4$^+$ T cells expressing TNF alone, TNF and IFNG, as well as TNF, IFNG, and IL2 (Fig. 6f).

**NAM mediates direct and indirect immunostimulatory effects.** To circumvent the heterogeneity of the M/D-driven model and acquire reliable mechanistic insights into the immunological circuitries that underlie the therapeutic activity of NAM, we performed depletion/neutralization experiments. This approach confirmed that the therapeutic activity of NAM depends on the immune system, as demonstrated by experiments involving the co-depletion of CD4$^+$ and CD8$^+$ T cells, lymphocyte subsets expressing NK1.1 (NK cells, NKT cells and a subset of activated CD8$^+$ T cells)[31], or IFNG neutralization (Fig. 7a–c). Similarly, the ability of NAM to counteract the growth of AT3 tumors was abolished by the co-depletion of CD4$^+$ and CD8$^+$ T cells as well as by the neutralization of IFNAR1, IFNG, and IL17 (Supplementary Fig. 4d–g).

M/D-driven carcinomas resemble human HR$^+$ BCs also in their limited sensitivity to PD-1 blockers[8] (Fig. 7d). In this setting, the therapeutic activity of NAM was superior to that of PD-1 blockers administered alone, and addition of PD-1 blockers to NAM treatment failed to ameliorate the therapeutic activity of the latter (Fig. 7d). In contrast, NAM synergized with PD-1 blockade in the TSA model (Fig. 7e). Similarly, NAM could be favorably combined with MTX-based chemotherapy in both the M/D-driven and AT3 models, and extended the survival of mice more than did either NAM or MTX alone (Supplementary Figs. 3B and 4H).

Finally, we performed single-cell RNAseq (scRNAseq) on CD45$^+$ leukocytes infiltrating untreated vs NAM-treated TSA tumors, revealing mild numerical alterations in the immune infiltrate of NAM-treated tumors, including an increased frequency of T cells and monocytes (including DCs) coupled to decreased abundance of tumor-associated macrophages (TAMs) and NK cells (Fig. 8a). More importantly, NAM provoked considerable shifts in the transcriptional profile of all these four immune cell populations (differentially expressed genes: 1884 in macrophages, 1541 in monocytes, 590 in T cells, and 283 in NK cells), including multiple alterations supporting innate and

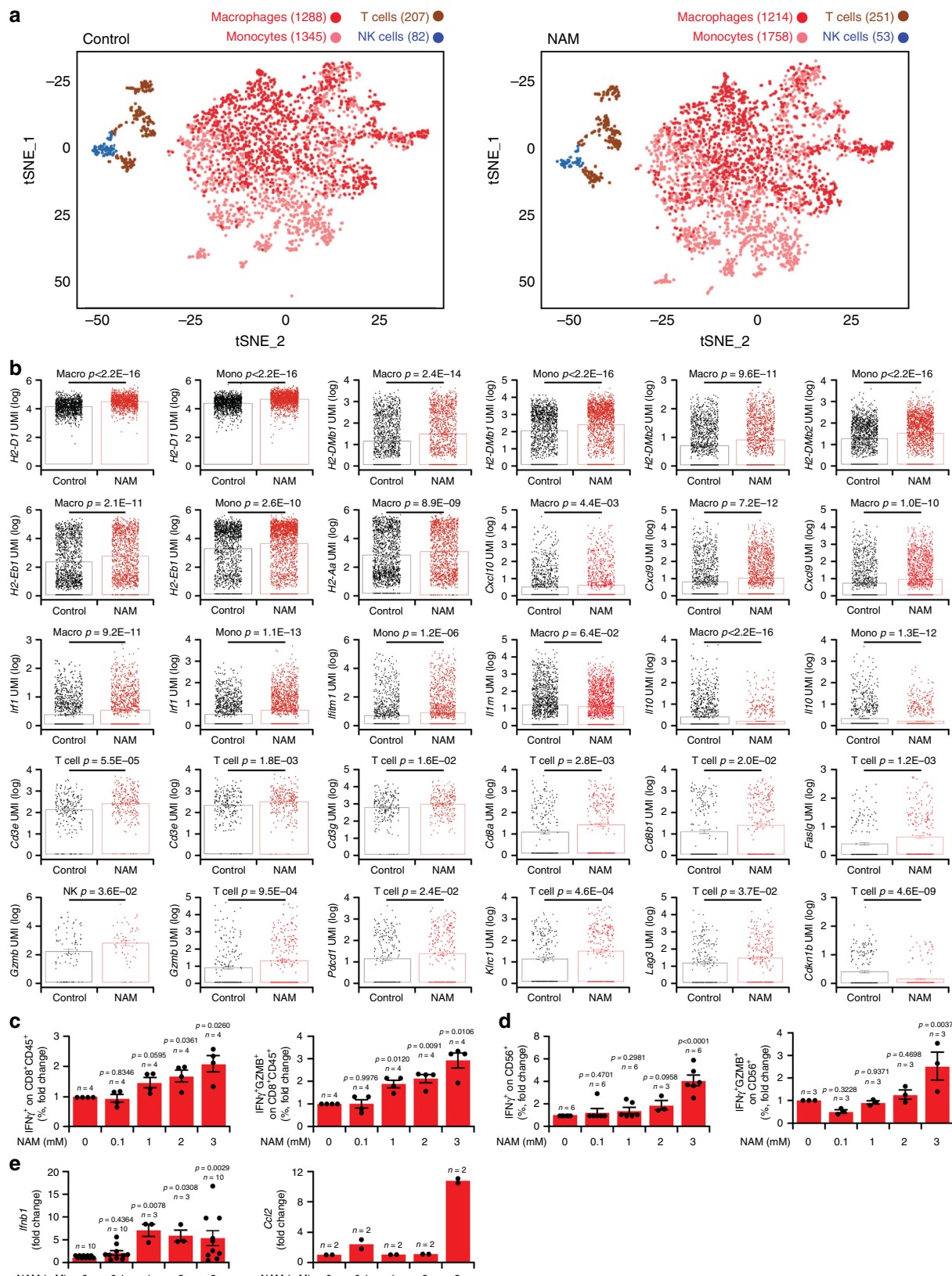

adaptive immunity (Supplementary Fig. 5a–d and Supplementary Data 1). In particular, monocytes and TAMs expressed increased levels of (1) multiple MHC molecules including H2-Aa, H2-DMb1, H2-DMb2, H2-D1, and H2-Eb1; (2) T-cell recruiting cytokines such as C-X-C motif chemokine ligand 9 (CXCL9) and

CXCL10, and (3) proteins responding to type I IFN signaling, such as interferon regulatory factor 1 (IRF1) and interferon induced transmembrane protein 1 (IFITM1) (Fig. 8b and Supplementary Data 1). Conversely, monocytes and TAMs infiltrating NAM-treated tumors exhibited lower expression of

**Fig. 8 NAM-treated tumors exhibit improved antigen presentation and superior cytotoxic functions. a** tSNE plots of untreated and NAM-treated TSA tumors. Number of cells in each of the main four populations is reported. **b** Differential expression of genes involved in immune regulation in CD45$^+$ cells isolated from untreated vs NAM-treated TSA tumors. Results are mean ± SEM plus individual data points. Number of cells independently analyzed for each gene is reported in (**a**), $p$ values (two-sided Wilcoxon test) are indicated. See also Supplementary Data 1. **c, d** Percentage of IFNG$^+$ and IFNG$^+$GZMB$^+$ amongst CD8$^+$ T cells (**c**) and CD56$^+$ NK cells (**d**) from peripheral blood mononuclear cells (PBMCs) of healthy donors subjected to non-specific activation overnight in the presence of the indicated concentrations of NAM. Results are means ± SEM plus individual data points. Number of biologically independent samples and $p$ values (one way-ANOVA plus Fisher LSD, as compared to untreated cells) are reported. **e** Relative expression levels of *Ifnb1* and *Ccl2* in TSA cells cultured in control conditions or exposed to the indicated concentrations of NAM for 48 h. Results are means ± SEM plus individual data points. Number of biologically independent samples and $p$ values (one way-ANOVA plus Fisher LSD, as compared to untreated cells) are reported.

immunosuppressive factors like IL10 and interleukin 1 receptor antagonist (IL1RN) (Fig. 8b and Supplementary Data 1). Amongst other changes, T cells infiltrating NAM-treated tumors exhibited (1) increased expression of genes involved in antigen recognition, including *Cd3d*, *Cd3e*, *Cd3g*, *Cd8a* and *Cd8b1*; (2) upregulation of effector molecules including granzyme B (GZMB) and Fas ligand (FASLG); (3) increased levels of surface activation markers like PD-1, killer cell lectin-like receptor subfamily C, member 1 (KLRC1, best known as NKG2A), and lymphocyte activating 3 (LAG3), and (4) reduced levels of negative cell cycle regulators, such as cyclin dependent kinase inhibitor 1B (CDKN1B) (Fig. 8b and Supplementary Data 1). Despite their limited number, NK cells infiltrating NAM-treated tumors also exhibited higher levels of GZMB as compared to NK cells from untreated tumors (Fig. 8b and Supplementary Data 1), which is indicative of (at least some degree of) functional activation. In line with these transcriptional changes, Gene Ontology (Supplementary Fig. 6b) and Hallmarks (Supplementary Fig. 6b) analysis revealed an enrichment of multiple gene sets linked to immune activation in the immune infiltrate of NAM-treated vs untreated TSA tumors, taken as a whole (CD45$^+$ cells) as well as in its specific components (macrophage, monocytes, T cells, NK cells). Moreover, immunofluorescence microscopy and RT-PCR corroborated the ability of NAM to increase the abundance of immune effector cells (CD8$^+$ T cells) and transcripts (*Gzmb*, *Ifng* and *Prf1*, although the latter sub-significantly) in the microenvironment of TSA and M/D-driven tumors (Supplementary Fig. 7a, b). Altogether, these findings lend additional support to the notion that NAM mediates therapeutic effects that depend on the immune system.

To obtain further mechanistic insights into this possibility, we exposed peripheral blood mononuclear cells (PBMCs) from healthy donors to phorbol 12-myristate 13-acetate (PMA) plus ionomycin as non-specific activation stimuli, alone or in the presence of NAM, finding that the latter favors the accumulation of both T cells and NK cells expressing IFNG alone or co-expressing IFNG and GZMB (Fig. 8c, d). We excluded the possibility that NAM could mediate robust cytotoxic effects on malignant cells, even at highly supraphysiological concentrations (Supplementary Fig. 7c), but identified the unsuspected ability of NAM to drive transcription from the *Ifnb1* locus, as well as from the *Ccl2* locus (Fig. 8e), which encodes a potent chemoattractant for monocytes. Altogether, these findings are in line with scRNAseq and efficacy data, and suggest that NAM mediates both direct and indirect immunostimulatory effects that impinge on T cell activation.

## Discussion

Here, we demonstrate that M/D-driven tumors recapitulate key immunobiological features of human HR$^+$ BCs, which account for the majority of BC[1]. M/D-driven tumors emerge in immunocompetent hosts via an ER-dependent oncogenic process, and progress as they evade immunosurveillance, culminating with the establishment of mammary tumors with a luminal B (highly

proliferative HER2$^-$ER$^+$VIM$^-$) phenotype that fails to convert into basal (ER$^-$VIM$^+$) over time or upon treatment. Moreover, M/D-driven tumors developing in immunocompetent mice resemble human luminal B BCs as they exhibit common *PI3KCA* mutations[32], poor immune infiltration coupled to limited sensitivity to ICBs[8], and can be accelerated by nutritional interventions that have been epidemiologically linked to increased propensity for HR$^+$ BC in humans, such as a HFD[17]. One of the caveats of this model relates to the fact that while ER$^+$ BC is the most common form of BC in women, only a fraction of cases of ER$^+$ BC is linked to the use of progestins[11]. Moreover, the M/D-driven model develops in a rather heterogenous manner, which calls for the use large experimental groups in support of statistical power, and displays a luminal B phenotype, which is common to other mouse BC models including TSA and AT3 cells[28,29], as well as mice expressing MMTV-PyMT[22]. That said, heterogeneity is also a feature of human ER$^+$ BC and to the best of our knowledge, no other model of HR$^+$ BC enabling to investigate oncogenesis, tumor progression and response to treatment in immunologically intact hosts has been documented[9]. Thus, M/D-driven tumors stand out as a privileged model for the study of BC immunosurveillance, resistance to (immuno)therapy, and sensitivity to nutritional interventions that operate by immunological mechanisms.

Interestingly, M/D-driven oncogenesis appears to be largely controlled by an immunological network involving interferon signaling and NK cell functions. This is at odds with other models of endogenous-driven carcinogenesis, such as methylcholanthrene (MCA)-dependent fibrosarcomas[33] as well as with numerous instances of spontaneous carcinogenesis that are accelerated by the lack of T cells or their molecular effectors[34], potentially constituting a peculiarity of HR$^+$ BCs. In this setting, NAM supplementation with the drinking water not only delays the manifestation of M/D-driven tumors when provided as a prophylactic intervention, but also mediates therapeutic effects when initiated at tumor detection. In both these settings, the effects of NAM depend on CD8$^+$ T lymphocytes but less so on NK cells. These findings raise the interesting possibilities that M/D-driven carcinogenesis in mice (and perhaps HR$^+$ mammary carcinogenesis in women) may be under natural immunosurveillance by NK cells[35], and that NAM can be harnessed to re-enable T cell-dependent tumor control in both prophylactic and therapeutic settings. Thus, established HR$^+$ tumors may exhibit a dysfunctional NK cell compartment potentially sensitive to reactivation by recently developed antibodies specific for NKG2A[36]. Further supporting this notion, while MCA-driven fibrosarcomas developing in immunocompetent animals are generally transplantable into both immunocompetent and immunodeficient hosts[33], the same does not apply to M/D-driven tumors, which are usually rejected by immunocompetent animals, irrespective of the immunological competence of the host in which they originally developed. This suggests that evolving M/D-driven tumors are poorly edited by T lymphocytes even in immunocompetent hosts, and hence evolve by accumulating

antigenic determinants that prevent engraftment in tumor-naïve immunocompetent mice. The precise molecular mechanisms underlying the inability of T cells to control M/D-driven carcinogenesis and edit evolving M/D-driven tumors remain to be elucidated.

In the course of this study, we discovered that NAM is particularly efficient at controlling the growth of various mouse models of luminal B BC, including M/D-driven tumors as well as TSA and AT3 tumors established in immunocompetent hosts, both as a standalone agent and in combination with clinically approved agents (e.g., anthracyclines and PD-1 blockers). The dose of NAM employed in this study (0.5% w/w with the drinking water) admittedly results in supraphysiological levels of blood-borne NAM (16.3 µg mL$^{-1}$). Although peak NAM concentrations of ~2.3 µg mL$^{-1}$ have been documented in healthy volunteers receiving 2 g nicotinic acid (a precursor of NAM) per os[20], whether higher steady-state NAM levels can be safely achieved in humans remains unexplored. That said, NAM has been successfully used in in randomized clinical trials to boost the efficacy of radiation therapy in patients with bladder and laryngeal carcinoma[37,38], as well as for the prevention of actinic keratosis and non-melanoma skin cancers[39]. The latter effect has been linked with reduced tumor infiltration by CD68$^+$ inflammatory macrophages[40], which is in line with our observations in the therapeutic setting. In a preclinical model of benzo(a)pyrene-induced lung carcinogenesis, oral (but not nasal) administration of NAM also demonstrated oncopreventive effects[41]. Thus, NAM might exert a relative broad oncosuppressive action, contrasting with other vitamins such as pyridoxine and calcitriol, which failed to enable oncoprevention in observational or interventional clinical studies[42–44]. Of note, haploidentical or mismatched related donor NK cells expanded in the presence of NAM are currently being investigated for the treatment of refractory/relapsed multiple myeloma and non-Hodgkin's lymphoma (NCT03019666), corroborating the elevated potential of NAM as an immunostimulant for clinical applications.

As for mode of action, it appears plausible that NAM acts as a therapeutic agent by re-establishing T cell-dependent immunosurveillance via both direct (T cell activation) and indirect (type I IFN and CCL2 secretion by cancer cells) mechanisms. However, the precise molecular cascade(s) underlying such beneficial effects of NAM remain(s) to be elucidated. Hydroxycarboxylic acid receptor 2 (HCAR2), a G protein-coupled receptor for nicotinic acid, has been reported to suppress mammary oncogenesis in mice by inhibiting cancer cell survival[45]. However, HCAR2 has a 1000-fold lower affinity for NAM than for nicotinic acid[46], and although some degree of NAM conversion into nicotinic acid by the gut microbiota cannot be excluded[47], hepatic metabolism efficiently removes nicotinic acid from the circulation by forming NAM[47]. Moreover, mouse BC cells exposed to highly supraphysiological NAM concentrations up to 80 mM did not undergo cell death considerably, ruling out cytotoxicity from the mechanisms through which NAM mediates therapeutic effects in our models (further corroborated by the inability of NAM to slow down tumor growth in vivo upon T cell depletion).

These observations not only cast doubts on the actual implication of HCAR2 in the beneficial effects of NAM against mammary tumors, but also argue against the involvement of sirtuin 1 (SIRT1) inhibition, which has been linked to the death of human BC cells exposed to high-dose NAM[48]. Of note, low NAM concentrations have been reported to activate (rather than inhibit) SIRT1 in numerous settings[49], and SIRT1 functions appear to be required for robust immune responses in a variety of settings[50,51]. Additional work is required to assess the impact of SIRT1 on the immunosurveillance of luminal B BC.

Finally, NAM might also act as an inhibitor of immunosuppressive NAD$^+$-consuming enzymes that generate NAM as a product of catalysis, such as CD38 (ref. [52]) or poly(ADP-ribose) polymerase 1 (PARP1). Intriguingly, PARP1 has recently been shown to limit type I IFN secretion by cancer cells with defects in the DNA damage response[53,54], and our findings indicate that NAM not only favors type I IFN secretion by TSA cells maintained in vitro, but also (1) elicits signatures of type I IFN signaling in immune cells infiltrating TSA tumors established in immunocompetent hosts, and (2) mediates prophylactic and therapeutic effects that depend (at least in part) on type I IFN responses. That said, whether PARP1 inhibition underlies the beneficial effects of NAM against mammary tumors remains to be experimentally verified.

Irrespective of these hitherto untested possibilities, NAM supplementation stands out as a safe and effective option for the prevention and treatment of HR$^+$ BC. Prospective, randomized clinical trials clarifying the preventive are therapeutic activity of NAM are urgently awaited.

## Methods

**Cell lines**. Mouse mammary adenocarcinoma TSA cells (#SCC177), mouse mammary carcinoma AT3 cells (#SCC178) and mouse fibrosarcoma MCA205 cells (#SCC173) were obtained from Millipore Sigma. All cell lines were cultured at 37 °C under 5% of $CO_2$, in the appropriate medium containing 10% fetal bovine serum (FBS), 100 U mL$^{-1}$ penicillin sodium and 100 µg mL$^{-1}$ streptomycin sulfate (both included in #15070063 from Thermo Fisher). TSA cells were cultured in Dulbecco's Modification of Eagle's Medium (DMEM, #11960044 from Thermo Fisher), AT3 cells in RPMI 1640 medium (#11875119, Thermo Fisher), and MCA205 cells in RPMI 1640 medium supplemented as above plus 1 mM sodium pyruvate (#11360070, Thermo Fisher) and 1 mM HEPES buffer (#15630106, Thermo Fisher). All cell lines were routinely checked for *Mycoplasma spp.* contamination by the PCR-based LookOut® Mycoplasma PCR Detection Kit (#MP0035, Millipore Sigma).

**Primary cell cultures**. After sacrifice, cells from M/D-driven tumors developed in WT or $Rag2^{-/-}$ mice were surgically recovered and washed in DMEM/F12 medium (#10565042, Thermo Fisher) supplemented with 100 U mL$^{-1}$ penicillin sodium, 100 µg mL$^{-1}$ streptomycin sulfate and 100 µg mL$^{-1}$ gentamycin. They were then dissociated in wash medium supplemented with 0.15% type A collagenase (#10103578001, Millipore Sigma) for 30 min at 37 °C (with occasional mechanical dispersion). After dissociation, cells were cultured in flasks pre-coated with 0.1% gelatin (4 h, 37 °C), in DMEM/F12 medium supplemented with 2% FBS, 50 µg mL$^{-1}$ gentamycin sulfate (#15750078, Thermo Fisher), 10 ng mL$^{-1}$ insulin (#I0516, Millipore Sigma) and 5 ng mL$^{-1}$ epithelial growth factor (EGF; #SRP3196 from Thermo Fisher).

**Mouse mammary gland organoids (MMGOs)**. For MMGO handling, plasticware was pre-coated with PBS supplemented with 2.5 mg mL$^{-1}$ bovine serum albumin (BSA, #700-100P from Gemini Bio-Products). Mammary glands were collected, and lymph nodes removed. Mammary glands were then minced with scissors and digested in DMEM/F12 medium supplemented with 5% FBS, 50 µg mL$^{-1}$ gentamicin sulfate, 5 ng mL$^{-1}$ insulin, 0.04% (w/v) Trypsin-EDTA (#15400054, Thermo Fisher) and 2 mg mL$^{-1}$ collagenase A (#10103586001, Roche) for 1 h at 37 °C at 100 rpm in a HulaMixer sample mixer (Life Technologies). Thereafter, mammary glands were subjected to vigorous shaking to disrupt adipocytes, and cells were pelleted at 520 g for 10 min. After further washing in DMEM/F12 medium, both the cell pellet and the fat layer were recovered. The latter was mechanically dissociated, pelleted again, and then added to the cell pellet. Thereafter, cells were resuspended for 5 min in DMEM/F12 medium supplemented with 4 U mL$^{-1}$ DNase I (#DPRF, Worthington Biochemical), pelleted, and cleared by differential centrifugation. Mammary organoids were finally collected and embedded in Matrigel® Matrix (#356255, Corning) for 3D culture.

**Mice**. WT or genetically modified C57BL/6 and BALB/c female mice (*Mus musculus*) of 6-15 weeks of age were employed. Mice were maintained in standard specific pathogen-free (SPF) housing conditions (20 ± 2 °C, 50 ± 5% humidity, 12h-12h light-dark cycles, food and water *ad libitum*), unless specified as per study design. Animal experiments followed the Federation of European Laboratory Animal Science Association (FELASA) guidelines, were in compliance with the EU Directive 63/2010 (protocol 2012_034A) and were approved by institutional ethical committees for animal experimentation at Gustave Roussy (no. 2016031417225217), Centre de Recherche des Cordeliers (no. 2016041518388910), and Weill Cornell Medical College (no. 2017-0007 and 2018-0002). WT C57BL/6 and BALB/c mice were obtained from Harlan France or Taconic Farms, $Rag2^{-/-}Il2rg^{-/-}$, $Rag2^{-/-}$, $Casp1^{-/-}$, $Il21^{-/-}$,

*Il21r*$^{-/-}$, *Il1r1*$^{-/-}$, *Nlrp3*$^{-/-}$, *Ifng*$^{-/-}$, and *Ifnar1*$^{-/-}$ mice were obtained from The Jackson Laboratory, *Il22*$^{-/-}$ and *Myd88*$^{-/-}$ mice were obtained from CNRS UMR 7355 (Orleans, France). MMTV-PyMT mice were obtained from Prof. M.F. Krummel (UCSF, San Francisco, CA, USA), *Becn1*$^{+/-}$ mice from Prof. B. Levine (UT Southwestern Medical Center, Dallas, TX, USA), *Atg5*$^{fl/fl}$ KRT5-Cre mice Prof. J. Penninger (Institute of Molecular Biotechnology, Vienna, Austria) and originally from Prof. Noboru Mizushima (University of Tokyo, Tokyo, Japan), and both ERα AF-2$^0$ and ER$^{C451A}$ mice were kindly provided by Prof. F. Lenfant (INSERM U1048, Toulouse, France). In all experiments, mice were routinely monitored for tumor growth and euthanatized when tumor surface reached 200-250 mm2 (ethical endpoint), or in the presence of overt signs of distress (e.g., hunching, anorexia, tumor ulceration).

**Oncogenesis**. Fifty mg slow-release (90 days) MPA pellets (#NP-161, Innovative Research of America) were implanted subcutaneously by surgery into 6–9-weeks-old female mice (day 0). Mice were administered 200 μL of a 5 mg mL$^{-1}$ 7,12-dimethylbenz[a]anthracene (DMBA; #D3254, from Millipore Sigma) solution in corn oil (#C8267, Millipore Sigma), by oral gavage once a week on weeks 1, 2, 3, 5, 6, and 7 after implantation of the MPA pellet.

**Transplantable breast cancers**. For tumorigenicity assays, $1.0 \times 10^6$ AT3, $0.1 \times 10^6$ TSA, $0.3 \times 10^6$ MCA205 cells, or $0.5 \times 10^6$ cells established from M/D-driven tumors evolving in *Rag2*$^{-/-}$ or C57BL/6 mice (MGT cells) were injected subcutaneously or in the mammary fat pad of syngeneic mice.

**Depletion/neutralization procedures**. All antibodies blocking or neutralizing specific immune cell populations, cytokines or cytokine receptors and their correspondic isotype controls were acquired from BioXCell (West Lebanon): CD4 (clone GK1.5, #BE0003-1), CD8α (clone 2.43, #BE0061), NKG2D (clone HMG2D, #BE0111), NK1.1 (clone PK136, #BE0036), PD-1 (clone RMP1-14, #BE0146), IFNγ (clone R4-6A2, #BE0054), IFNAR1 (clone MAR1-5A3, #BE0241), IL-17A (clone 17F3, #BE0173). Antibodies in the amount of 0.2-0.4 mg/mouse were administered intraperitoneally once every 8 days or, in the case of anti-PD1, every 3 days for 3 times. For vaccinations experiments, ANXA1 neutralization was achieved with a specific antibody (clone 29, #610066 from BD Biosciences).

**Food and drugs**. Tamoxifen (#T5648, Millipore Sigma) and NAM (#N0636, Millipore Sigma) were administered in the drinking water at concentrations of 10 μg mL$^{-1}$ and 0.5% (w/v), respectively. For starvation experiments, mice underwent biweekly 24-h-long fasting cycles (with water ad libitum). Pellets for NR supplementation (3.33 g Kg$^{-1}$ pellets), HSD and HFD, and the corresponding placebo pellets were obtained from Ssniff Spezialdiäten GmbH. Pyridoxine hydrochloride (187 mg Kg$^{-1}$), calcitriol (300 μg Kg$^{-1}$), and spermidine (50 mg Kg$^{-1}$) (all from Millipore Sigma, cat. #P9755, #C0225000 and # S2626, respectively) were delivered *i.p.* in 100 μL PBS once a week. For in vivo experiments, 5.17 mg Kg$^{-1}$ MTX (#M2305000, Millipore Sigma) was administered *i.p.* in 200 μL PBS once at randomization. Solutions and bottles were changed three times a week.

**Vaccination experiments**. Cell lines or MMGOs were exposed in vitro to 1 μM MTX or 20 Gy irradiation and incubated for 24 h to obtain around 50–60% of dying cells, followed by injection into the mammary fat pad of WT mice 1 week prior, in the 4th week of (break week), and 2 days after, DMBA gavage. ANXA1 blockade was achieved by injecting 12.5 μg of the blocking antibody (or the correspondent isotype control) in the mammary fat pad 24 h before and after vaccination, and by complementing the vaccine with 250 μg mL$^{-1}$ of the same antibody.

**Histological studies**. Tissues were fixed in 4% formaldehyde (#F8775, Millipore Sigma) overnight and kept in ethanol 70% until inclusion in paraffin. Tissue processing was done by the Electron Microscopy & Histology Service of Weill Cornell Medicine. Fixed tissue was processed and embedded using a Tissue-Tek VIP® 6 AI Vacuum Infiltration Processor and Tissue Embedding Console (Sakura). Processing consisted of multiple ethanol dehydration steps of 1.5 h each: 70, 85, 95, 95, 100, and 100%. This was followed by two exchanges of HistoChoice® Clearing Agent (#H2779-1L, Millipore Sigma) and three exchanges of paraffin (2 h each). All steps were performed using vacuum and pressure. The paraffin steps were performed at 60 °C, all other steps were performed at room temperature (RT). The blocks were sectioned at 7-μm thick, collected onto positively charged glass slides, and stained with Hematoxylin and Eosin Stain Kit (#H-3502, Vector Laboratories). The histology of M/D-driven tumors was compared with archival images of HR$^+$ BC tissues from Weill Cornell Medicine (kindly provided by Dr. Syed Hoda).

**Immunofluorescence microscopy—M/D-driven tumors**. Slides with M/D-driven tumors obtained for histological studies were deparaffinized and re-hydrated by three incubations (5 min each) in xylenes (#214736, Millipore Sigma) followed by serial (2×, 3 min each) incubations in 100, 90, 80, and 70% ethanol. Autofluorescence was blocked with the MaxBlock™ Autofluorescence Reducing Reagent (#MB-L, MaxVision Biosciences Inc.) for 10 min at RT, followed by wash in 60% ethanol and incubation in deionized water (3×, 5 min each). Antigen retrieval was achieved by boiling sections for 50 min in 1X Citrate Buffer (pH 6, #C9999, from

Millipore Sigma), followed by rinsing with PBS (4×) and blockage of non-specific binding sites with 3% BSA in TBS-Tween for 30 min at RT. Slides were then incubated overnight at 4 °C with the following primary antibodies: anti-ESR1-AlexaFluor488 (1:20, #sc-8005 from Santa Cruz Biotechnology) and anti-VIM (1:250, #GTX100619, from GeneTex). Next, slides were rinsed 3× in TBS-Tween, incubated with an AlexaFluor594-conjugated goat anti-rabbit antibody (1:500; #ab150084 from Abcam) for 30-60 min at RT, rinsed again 3× in TBS-Tween and 1× in deionized water, and incubated with Post-Detection Conditioner (part of #MB-L, MaxVision Biosciences Inc.) for 5 min at RT. Finally, slides were incubated in 5 μg mL$^{-1}$ Hoechst 33258 (#H3569, Thermo Fisher) for 10 min at RT, mounted with ProLong™ Diamond Antifade Mountant (#P36961, Thermo Fisher), and imaged on an Eclipse TiE Motorized Digital Fluorescence Microscope operated by NIS-Elements AR v. 4.11 (Nikon).

**Immunofluorescence microscopy—TSA tumors**. Freshly excised TSA tumors were fixed in zinc fixative (#552658, BD Biosciences) prior to embedding in paraffin. After deparaffinization and dehydration, slides were boiled in de-cloaker media (#RD913M, Zytomed), treated with peroxide block (# ZUC019-008, Zytomed), normal goat serum (#S-1000-20, Vector Laboratories) and stained with primary antibodies specific for CD8A (1:2000, #14-0808-80, from Thermo Fisher) and secondary ImmPRESSTM HRP anti-rat IgG antibodies (ready for use, #MP-7444-15, from Vector Laboratories). Detection was performed with the Opal650 OPAL reagent (1:300, part of #NEL811001KT, Akoya Biosciences). After nuclear counterstaining with 4′,6-diamidino-2-phenylindole (DAPI, 2 drops mL$^{-1}$, part of #NEL811001KT, from Akoya Biosciences), slides were mounted and evaluated on a Vectra Polaris™ Automated Quantitative Pathology Imaging System operated by embedded software Vectra Polaris v.1 (Perkin Elmer). Whole slides were scanned first, followed by acquisition of multiple regions of interest covering almost the whole tumor surface with Phenochart v. 1.0.8, which were analyzed with inForm v. 2.4.6 (Akoya Biosciences)[55,56].

**Cell death**. Cell death was quantified by flow cytometry on a MACSQuant analyzer operated by MACSQuantify™ v. 2.11 (Miltenyi) upon staining with 0.5 μg mL$^{-1}$ propidium iodide (PI, #P4170 from Millipore Sigma) staining[57]. Gating procedure is exemplified in Supplementary Fig. 8.

**RT-PCR**. Mouse *Ccl2* and *Ifnb1* levels were quantified relative to *Rpl13* levels by 2-steps RT-PCR based on commercial primer sets from Bio-Rad (*Ccl2*, unique assay ID qMmuCED0003785; *Ifnb1*, unique assay ID qMmuCED0002606; *Rpl13a*, unique assay ID qMmuCED0040629), the SuperScript™ VILO™ Master Mix (#11755500, from Thermo Fisher) and the iTaq Universal SYBR Green Supermix (#1725121, Bio-Rad). Alternatively, mouse *Gzmb*, *Ifnb1*, *Ifng* and, *Prf1* levels were quantified relative to *Rpl13* levels by 1-step RT-PCR based on commercial primer and probe sets (*Gzmb*, #Mm00442837_m1; *Ifnb1*, #Mm00439552_s1; *Ifng*, #Mm01168134; *Pfr1*, #Mm00812512_m1; *Rpl13*, #Mm02526700_g1) and the TaqMan™ Fast Virus 1-Step Master Mix (#4444434), all from Thermo Fisher. Amplifications were run on a 7500 RT-PCR system (Applied Biosystems) operated by embedded software v. 2.3 as per thermal protocols provided by the primer manufacturer. RT-PCR data was normalized according to the $\Delta C_t$ or $\Delta\Delta C_t$ methods.

**CRISPR/Cas9**. TSA cells were transfected with a control commercial CRISPR-cas9 plasmid (#CRISPR06-1EA, from Millipore Sigma) or with a CRISPR-cas9 plasmid specific for *Atg7* (custom-made by Millipore Sigma based on #CRISPR06-1EA), using the TransIT-CRISPR® reagent (#T1706, Sigma Aldrich). GFP$^+$ clones were sorted on a FACSAria II Sorter operated by FACSDiva™ v. 6.1.3 (from BD Biosciences) into 96-well plates, followed by clone selection and confirmation of ATG7 status by immunoblotting.

**ShRNA transfection**. AT3 cells were stably transfected with commercial plasmid encoding an *Atg7*-specific shRNA (#TG504956, Origene Technologies) using the FuGen® HD transfection reagent (#E2311, Promega), as per manufacturer instructions. ATG7 status was confirmed by immunoblotting.

**NAM quantification**. Targeted LC/MS analyses of NAM were performed on a Q Exactive™ Orbitrap Mass Spectrometer (Thermo Fisher) coupled to a Vanquish™ HPLC system operated by Xcalibur v. 4.0.27.19 (Thermo Fisher). A SeQuant® ZIC®-HILIC column (2.1 mm i.d. × 150 mm, Merck) was used for metabolite separation. Flow rate was 150 μL/min. Buffers consisted of 100% acetonitrile for mobile A, and 0.1% NH$_4$OH/20 mM CH$_3$COONH$_4$ in water for mobile B. Gradient ran from 85% to 30% A in 20 min followed by a wash with 30% A and re-equilibration at 85% A. NAM was identified in positive ion mode on the basis of exact mass within 5 ppm and standard retention time. Absolute quantitation was performed using a NAM-based standard curve.

**Immunoblotting**. For detection of autophagy biomarkers in vivo, tissues were snap frozen, and then mechanically disrupted before cell lysis. For ATG7 detection, cell lines were subjected to standard lysis procedures. In both cases, 50 μg proteins were

separated on NuPAGE® Novex® Bis-Tris 4–12% pre-cast gels (Invitrogen) and electrotransferred to polyvinyldifluoride (PVDF) membranes (Millipore Sigma). Membranes were blocked with 0.05% Tween 20 (v/v in TBS) supplemented with 5% non-fat powdered milk for 1 h and incubated overnight with primary antibody specific for MAP1LC3B (1:1000, #2775 from Cell Signaling Technology), SQSTM1 (1:1000, #5114 from Cell Signaling Technology), ATG7 (1:1000, clone D12B11, #8558 from Cell Signaling Technology; or 1:3000, clone ATG7-13, #SAB4200304 from Millipore Sigma), or ACTB (1:1000, clone 13E5, #4970 from Cell Signaling Technology; or 1:2000, clone 8H10D10, #3700 from Cell Signaling Technology), at 4 °C. Primary antibodies were detected with horseradish peroxidase (HRP)-conjugated anti-mouse (#NA931, from GE Healthcare Life Sciences, 1:5000) or anti-rabbit (#NA934, from GE Healthcare Life Sciences, 1:5000) secondary antibodies and revealed with the Pierce™ ECL Plus chemiluminescent substrate (#32132, Thermo Fisher) on a C600 Gel Doc & Western Imaging System operated by cSeries Capture v. 1.6.8.1110 (Azure Biosystems).

**Flow cytometry—Tumor and mammary gland processing**. M/D-driven tumors and mammary glands were recovered, cut with scissors and digested in RPMI medium supplemented with 26.67 µg mL$^{-1}$ Liberase™ (#5401119001, Millipore Sigma) and 0.0167 MU mL$^{-1}$ DNase I for 30 min at 37 °C. RPMI medium supplemented with 10% FBS was then added to stop enzymes activity, and tumors were crushed on a 100 µm cell strainer with the back of a syringe plunger. After washing, cells were pelleted for 5 min at 300 g and resuspended in 10 mL RPMI medium supplemented with 10% FBS.

**Flow cytometry—spleen and lymph node processing**. Spleens and inguinal lymph nodes were collected and crushed between two microscope glass slides. Cells were resuspended in RPMI medium, filtered through 70 µM MACS® SmartStrainers (#130-098-462, Miltenyi) and pelleted for 5 min at 300 g. Splenocytes and lymph node cells were finally resuspended in 2 mL and 500 µL of RPMI supplemented with 10% FBS, respectively.

**Flow cytometry—in vitro T and NK cell phenotyping**. PBMCs were isolated from a 10 mL blood aliquot from healthy donors (following ethical guidelines of the University Hospital Motol, Prague, and upon collection of informed consent forms) by Ficoll® Paque PLUS (#17-1440-02, GE Healthcare Life Sciences) gradient centrifugation. PBMCs were then cultured in the presence of 50 ng mL$^{-1}$ phorbol 12-myristate 13-acetate (PMA, # P8139 from Millipore Sigma) plus 1 µg mL$^{-1}$ ionomycin (#I3909, Millipore Sigma) alone or combined with NAM, followed by 3-h incubation with brefeldin A (1:1000, # 420601 from BioLegend). Cells were then washed in PBS, stained with either anti-CD45-PE (1:15, clone HI30, #MHCD4517 from Thermo Fisher) and anti-CD3-AlexaFluor700 (1:20, clone MEM-57, #A7-202-T100 from EXBIO) and anti-CD8-HV500 (1:20, clone RPA-T8, #560775 from BD Biosciences) monoclonal antibodies, or anti-CD45-V500 (1:15, clone HI30, #560777 from BD Biosciences), anti-CD3-AlexaFluor700 (1:20, clone OKT3, # A7-631-T100 from EXBIO), and anti-CD56-ECD (1:15, clone N901, #B49214 from Beckman Coulter) monoclonal antibodies, then fixed in eBioscience™ fixation/permeabilization buffer (part of #88-8824-00, Thermo Fisher), further permeabilized with eBioscience™ permeabilization buffer (part of #88-8824-00, Thermo Fisher) and intracellularly stained with anti-IFNγ-PE-Cy7 (1:100, clone 4S.B3, #25-7319-82 from Thermo Fisher) and anti-GZMB-BV421 (1:25, clone GB11, #563389 from BD Biosciences) monoclonal antibodies. Flow cytometry was performed on a LSRFortessa™ Flow Cytometer operated by FACSDiva™ v. 6.2 (BD Biosciences), and data were analyzed with FlowJo v. 9.5.3 (TreeStar, Inc.). Gating procedure is exemplified in Supplementary Fig. 8.

**Flow cytometry—phenotyping of the immune infiltrate**. For immune cell phenotyping, 100 µL of spleen, lymph node, tumor, and mammary gland suspensions underwent one of the following procedures, after which stained samples were acquired on an LSR II Flow Cytometer operated by FACSDiva™ v. 6.1.3 (BD Biosciences) and analyzed with FlowJo v. X.6.2.

To assess cytokine production by T lymphocytes, cells were (re-)stimulated for 5 h in serum-free CTL-Test™ PLUS Medium (#CTLT-010, ImmunoSpot) containing 20 ng mL$^{-1}$ PMA and 1 µg mL$^{-1}$ ionomycin together with BD GolgiPlug™ (1:100, #555029 from BD Biosciences). Afterwards, T cells were stained with LIVE/DEAD™ Fixable Yellow dye (1:500, #L34959 from Thermo Fisher). Fc receptors were blocked with the anti-mouse CD16/CD32 reagent Mouse BD Fc Block™ (1:200, clone 2.4G2, #553141 from BD Biosciences). Staining of surface markers was performed with the following fluorochrome-conjugated antibodies: anti-CD3-BV421 (1:100, clone 145-2C11, #562600 from BD Biosciences), anti-CD8-FITC (1:400, clone 53-6.7, #553030 from BD Biosciences), and anti-CD4-PerCP-Cy5.5 (1;400, clone RM4-5, #45-0042-82 from Thermo Fisher). Cells were then fixed and permeabilized in Cytofix/Cytoperm™ buffer (#554714, BD Biosciences), and intracellular cytokine staining was performed with anti-IFN-γ-APC (1:100, clone XMG1.2, #554413 from BD Biosciences), anti-TNFα-APC-Cy7 (1:200, clone MP6-XT22, #506307 from BioLegend), and anti-IL-2-PE (1:300, clone JES6-%H4, #554428 from BD Biosciences). Gating procedure is exemplified in Supplementary Fig. 8.

To measure the populations of αβ T lymphocytes and their activation/exhaustion status, cells were stained with LIVE/DEAD™ Fixable Yellow dye and Fc receptors were blocked as described above. Then, a cell surface staining was realized with the following fluorescent antibodies: anti-CD3-APC (1:200, clone 17A2, #17-0032-82 from Thermo Fisher), anti-CD8-PE (1:400, clone 53-6.7, #553032 from BD Biosciences), anti-CD4-PerCP-Cy5.5 (1:400, clone RM4-5, #45-0042-82 from Thermo Fisher), anti-CD25 PE-Cy7 (1:100, clone PC61.5, #25-0251-82 from Thermo Fisher), and anti-PD-1-APC/Fire750 (1:100, clone 29 F.1A12, #135239 from BioLegend). Cells were then fixed and permeabilized in 1X Foxp3 Transcription Factor Staining Buffer (part of #A25866A, from Thermo Fisher). Finally, an intranuclear staining was performed with anti-FOXP3-FITC (1:50, clone FJK-16s, #11-5773-82 from Thermo Fisher) and anti-Ki67-AlexaFluor700 (1:100, clone B56, #561277 from BD Biosciences). Gating procedure is exemplified in Supplementary Fig. 8.

Alternatively, cells were stained with LIVE/DEAD™ Fixable Yellow dye and Fc receptors were blocked as described above, then fixed in Cytofix/Cytoperm™ buffer after surface staining with anti-CD3-FITC (1:800, clone 17A2, #11-0032-82 from Thermo Fisher), anti-NK1.1-PerCP-Cy5.5 (1:100, clone PK136, #551114 from BD Biosciences), anti-B220-V450 (1:100, clone RA3-6B2, # 560473 from BD Biosciences) and anti-CD19-APC-Vio770 (1:50, clone REA749, #130-111-886 from Miltenyi). Gating procedure is exemplified in Supplementary Fig. 8.

**RNASeq**. Tissues of interest (including tumors) were recovered, stabilized in RNAlater RNA Stabilization Reagent (#R0901, Millipore Sigma) and RNA was extracted using RNeasy Mini kits (#74104, Qiagen), as per manufacturer's instructions. Sequencing, data quality, reads repartition (e.g., for potential ribosomal contamination) was performed by GenoSplice technology (www.genosplice.com) using Mouse Gene v. 1.0 ST Array for data presented in Fig. 1, or FastQC v. 0.11.2, Picard-Tools v. 1.119, Samtools v. 1.0, and RSeQC v. 2.3.9 for data presented in Fig. 6 and Supplementary Fig. 3.

**Single-cell RNAseq—cell isolation**. TSA tumors were recovered from three mice per group, and dissociated using the Tumor Dissociation Kit, mouse (#130-096-730, Miltenyi) and gentleMACS™ Octo Dissociator (Miltenyi), following manufacturer's instructions. Upon pooling, samples were subjected to magnetic separation of CD45$^+$ cells using CD45 MicroBeads (#130-052-301, Miltenyi) and a MACS Separator (Miltenyi), as per manufacturer's instructions. Viability was tested upon staining a cell aliquot with 0.05% trypan blue (#T6146, Millipore Sigma) in PBS on a Cellometer AutoT4 cell counter (Nexcelom Bioscience), and CD45$^+$ cells were resuspended in PBS supplemented with 0.1% BSA.

**Single-cell RNAseq—sample preparation and processing**. Library preparation for Single Cell 3′ RNA-seq v.2, sequencing and post-processing of the raw data was performed at the Epigenomics Core at Weill Cornell Medicine, as follows. Single-cell RNAseq libraries were prepared according to 10X Genomics specifications (Single Cell 3′ Reagent Kits v.2 User Guide PN-120236, 10x Genomics). Two independent cell suspensions (~60% viable) at a concentration of ~900 cells µL$^{-1}$ were loaded onto to the 10X Genomics Chromium platform to generate barcoded single-cell GEMs, targeting about 3000 single cells per sample. GEM reverse transcription (53 °C for 45 min, 85 °C for 5 min; held at 4 °C) was performed in a C1000 Touch Thermal cycler with 96-Deep Well Reaction Module (Bio-Rad). After reverse transcription, GEMs were ruptured and single-stranded cDNA was cleaned up with DynaBeads MyOne Silane Beads (#37002D, Thermo Fisher). cDNA was amplified for 12 cycles (98 °C for 3 min; 98 °C for 15 s, 67 °C for 20 s, 72 °C for 1 min) using the C1000 Touch Thermal cycler with 96-Deep Well Reaction Module. Quality of the cDNA was assessed using a Bioanalyzer 2100 (Agilent), obtaining a product of about 1693bp. This cDNA was enzymatically fragmented, end-repaired, A-tailed, subjected to a double-sided size selection with SPRIselect beads (#B23317, Beckman Coulter) and ligated to adaptors provided in the kit. A unique sample index for each library was introduced through 13 cycles of PCR amplification using the indexes provided in the kit (98 °C for 45 s; 98 °C for 20 s, 54 °C for 30 s, and 72 °C for 20 s × 14 cycles; 72 °C for 1 min; held at 4 °C). Indexed libraries were subjected to a second double-sided size selection, and libraries were then quantified using on a Qubit 4 Fluorometer (Thermo Fisher). Quality was assessed on a Bioanalyzer 2100, obtaining an average library size of 455 bp. Libraries were finally diluted to 2 nM and clustered on a HiSeq2500 System (Illumina) on high output mode at 10 pM on a pair-end read flow cell and sequenced for 26 cycles on R1 (10X barcode and the UMIs), followed by 8 cycles of I7 Index (sample Index), and 98 bases on R2 (transcript), with a coverage around 100 M reads per sample. Primary processing of sequencing images was done using Illumina's Real Time Analysis (RTA) software v. 3.4.4.

**Reproducibility and data management**. Unless otherwise stated, all experiments have been conducted in two independent instances with similar results. In each individual experiment, samples were measured once. Transcriptomic and scRNAseq findings are from a single experiment involving the indicated number of mice. Unless otherwise specified, Excel 2013 (Microsoft) and Prism v. 8.4 (GraphPad) were used for data management, graphing and statistical analyses.

Illustrator 2020 (Adobe) and Photoshop 2020 (Adobe) were used for figure preparation.

**Statistical analysis—tumor incidence and growth.** Tumor surface was routinely evaluated by the ellipse area formula: $(\pi \times A \times B)/4$, where $A$ and $B$ are the largest and smallest lesion diameter, respectively. Statistical significance on growth curves was assessed by the TumGrowth software v. 1[58] or two-way ANOVA corrected for row (time) and column (treatment) factors. Statistical significance on Kaplan Meyer curves (TFS, OS and TTD) was assessed by two-sided log-rank (Cox proportional hazards model) or one way-ANOVA plus Fisher LSD test.

**Statistical analysis—in vitro, imaging and flow cytometry assays.** Statistical significance on RT-PCR, immunofluorescence microscopy, and flow cytometry (including cell death) data were assessed by one way-ANOVA plus Fisher LSD test for comparisons involving more than two groups of samples or unpaired, two-sided Student's t test for comparisons involving only two groups of samples.

**Statistical analysis—bulk RNAseq.** For data presented in Fig. 1, we used the Affymetrix© Mouse Gene v. 1.0 ST Array. Data were normalized using Affymetrix© TAC v. 4.0.1. Analyses was performed within the R computational environment v. 3.4. Specifically, genes were selected according to their differential expression by using limma v. 3.32.10 (ref. [59]), using as thresholds |logFC| > 2 and p value < 0.05. GO enrichment was analyzed using the DAVID Functional Annotation Tool v. 6.8 (ref. [60]) on differentially expressed genes. For comparing mouse RNAseq data with gene expression in human BC (data presented in Supplementary Fig. 1), we harnessed microarray data from the TCGA database. In the R computational environment v. 3.6.0, we renormalized mouse data (centered and reduced) in order to obtain similar distribution as human data. We then applied the function subtype.cluster of genefu v. 2.16.0 (ref. [61]) based on previously published BC classification algorithms[62,63]. Other classification methods of the package genefu produced similar results. Heatmap with hierarchical clustering analysis was done using ComplexHeatmap v. 2.0.0 based on the Euclidean distance and ward.D2 clustering method. ENSEMBL Genes 100 was used to identified orthologous genes between human and mouse.

For data presented in Fig. 6 and Supplementary Fig. 3, analysis was performed by GenoSplice technology (www.genosplice.com). Reads were mapped using STAR v. 2.4.0 (ref. [64]) on the mm10 mouse genome assembly, followed by gene expression analysis[65]. In brief, for each gene present in Mouse FAST DB v. 2016_1 annotations, reads aligning on constitutive regions (that are not prone to alternative splicing) were counted. Based on these counts, normalization and differential gene expression were performed using DESeq2 v. 1.28.1 (ref. [66]) on the R computational environment (v.3.2.5). Only genes expressed in at least one of experimental conditions were analyzed further. To this aim, genes were considered as expressed if their rpkm value was greater than 93% of the background rpkm value based on intergenic regions. Results were considered statistically significant for p values ≤ 0.05 and fold changes ≥ 1.5. Gene modules were defined based on co-expression of genes within a given module. All genes within a given module had to be correlated to each other. A minimum Pearson correlation coefficient of 0.7 (i.e., only positive correlation) was required and a minimum of 50 genes was used to define a module. Clustering and heatmaps were obtained using the dist and hclust functions of R, based on the Euclidean distance and Ward agglomeration method. Analysis for enriched GO terms, KEGG pathways and REACTOME pathways were performed using the DAVID Functional Annotation Tool v. 6.8. GO terms and pathways were considered enriched if fold enrichment ≥ 2.0, uncorrected p value ≤ 0.05, and minimum number of regulated genes in pathway/term ≥ 2.0. Analysis was performed on all regulated genes, upregulated genes only, and downregulated genes only. A synthesis of these analyses was made to provide a single list of results. Analysis of the enriched GO pathways for commonly upregulated genes on both AT3 and M/D-driven tumors was performed using Ingenuity® Pathway Analysis (IPA®) v. 52912811.

**Statistical analysis—single-cell RNASeq.** The 10X Genomics Cell Ranger Single Cell Software Suite v2.1.0 was used to perform sample de-multiplexing, alignment (based on the mm10 mouse genome assembly), filtering, UMI counting, single-cell 3′end gene counting and performing quality control, using the manufacturer parameters. Unsupervised cell clustering was carried out using Seurat v. 2.3.4 (ref. [67]) in the R computational environment v. 3.5.3. Cells with <1000 genes or <3000 UMIs or >10% mitochondrial genes and genes detected in <3 cells were excluded from the analysis. Gene expression raw counts were normalized following a global-scaling normalization method with a scale factor of 10,000 and a log transformation, using the Seurat NormalizeData function. The top 2775 highly variable genes were selected, followed by principal component analysis (PCA). Top 35 PCs were further used on harmony v. 0.1.0 (ref. [68]) and remove the effect of confounding factors between two samples. The first 35 harmony results were used to generate two-dimensional t-Distributed Stochastic Neighbor Embedding (RunTSNE in Seurat, perplexity = 30) and unsupervised cell clustering by a shared nearest neighbor (FindClusters in Seurat, k.param = 30 and resolution = 0.6). A list of conserved cell-type specific genes was generated by FindAllMarkers function in Seurat. User defined genes were employed to identify cell types and confirmed by SingleR v. 0.2.2 (ref. [69]). Gene set enrichment analysis and bar charts were generated using ggpubr v. 0.2 and ggplot2 v. 3.1.1. Statistical assessments on differential gene expression and enrichment analysis were based on the MAST model[70] and fast gene set enrichment analysis (FGSEA)[71], respectively. Statistical significance on individual genes was performed by two-sided Wilcoxon test.

**Reporting summary.** Further information on research design is available in the Nature Research Reporting Summary linked to this article.

## Data availability

Transcriptomic data generated in the context of this study have deposited at Gene Expression Omnibus (GEO) and are publicly available under accession numbers GSE150921, GSE150966, GSE150967 and GSE151197. The Cancer Genome Atlas (TCGA) breast cancer (BRCA) dataset is publicly available at https://xenabrowser.net/datapages/?cohort=GDC%20TCGA%20Breast%20Cancer%20(BRCA)&removeHub=https%3A%2F%2Fxena.treehouse.gi.ucsc.edu%3A443. The mm10 mouse genome assembly and ENSEMBL Genes 100 can be accessed at https://useast.ensembl.org/Mus_musculus/Info/Index and https://www.ensembl.org/biomart/martview/eb26fab860902330900c3b118e3cd906, respectively. GO terms, KEGG pathways, REACTOME pathways and Hallmarks terms are freely available to academic users at http://geneontology.org/docs/downloads/, https://www.kegg.jp/kegg/download/, https://reactome.org/download-data, and https://www.gsea-msigdb.org/gsea/msigdb/collections.jsp#H, respectively. Additional data supporting the findings of this study are available from the corresponding authors upon reasonable request. A reporting summary for this article is available as a Supplementary Information File. Unique materials used for this research are available from the corresponding authors upon reasonable request. Source data are provided with this paper.

## Code availability

Custom code used for this research has been deposited at GitHub and is available at https://github.com/nyuhuang/scRNAseq-LorenzoBlood, and https://github.com/tonedivad/TumGrowth, or from the corresponding authors upon reasonable request.

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

## Acknowledgements

We are grateful to F. Lenfant (INSERM U1048, Toulouse, France) and J. Penninger (Institute of Molecular Biotechnology, Vienna, Austria) for kindly providing us with mice bearing genetic alterations in *Esr1* and *Atg5*, respectively. We are indebted to B. Levine (UT Southwestern, Dallas, TX, USA) and Dr. M.F. Krummel (UCSF, San Francisco, CA, USA) for kindly providing us with *Becn1*$^{+/-}$ and MMTV-PyMT mice, respectively. We thank Genom'IC (Cochin Institute, Paris, France) and GenoSplice (Paris, France) for help with RNA-Seq analysis. We thank S. Hoda (Weill Cornell Medicine, New York, NY, US) for providing human breast cancer microphotographs, G. Inghirami (Weill Cornell Medicine, New York, NY, US) for pathology assessments, G. Petroni (Weill Cornell Medicine, New York, NY, US) for critical reading and help with the finalization of the paper, and the Proteomics and Metabolomics Core Facility of Weill Cornell Medicine (New York, NY, US) for NAM quantification. M.P.L. is supported by Fondation pour la Recherche Médicale (FRM FDT201904008383). J.H. is supported by the Foundation Philanthropia. B.S. is supported from grants by the German Cancer Aid (#70113311 and #34102524). G.K. is supported by the Ligue contre le Cancer (équipe labellisée); Agence National de la Recherche (ANR) – Projets blancs; ANR under the frame of E-Rare-2, the ERA-Net for Research on Rare Diseases; Association pour la recherche sur le cancer (ARC); Cancéropôle Ile-de-France; Chancelerie des universités de Paris (Legs Poix), Fondation pour la Recherche Médicale (FRM); a donation by Elior; European Research Area Network on Cardiovascular Diseases (ERA-CVD, MINO-TAUR); Gustave Roussy Odyssea, the European Union Horizon 2020 Project Oncobiome; Fondation Carrefour; High-end Foreign Expert Program in China (GDW20171100085 and GDW20181100051), Institut National du Cancer (INCa); Inserm (HTE); Institut Universitaire de France; LeDucq Foundation; the LabEx Immuno-Oncology; the RHU Torino Lumière; the Seerave Foundation; the SIRIC Stratified Oncology Cell DNA Repair and Tumor Immune Elimination (SOCRATE); and the SIRIC Cancer Research and Personalized Medicine (CARPEM). The LG lab is supported by a Breakthrough Level 2 grant from the US Department of Defense (DoD), Breast Cancer Research Program (BRCP) (#BC180476P1), by the 2019 Laura Ziskin Prize in Translational Research (#ZP-6177, PI: Formenti) from the Stand Up to Cancer (SU2C), by a Mantle Cell Lymphoma Research Initiative (MCL-RI, PI: Chen-Kiang) grant from the Leukemia and Lymphoma Society (LLS), by a startup grant from the Dept. of Radiation Oncology at Weill Cornell Medicine (New York, US), by a Rapid Response Grant from the Functional Genomics Initiative (New York, US), by industrial collaborations with Lytix (Oslo, Norway) and Phosplatin (New York, US), and by donations from Phosplatin (New York, US), the Luke Heller TECPR2 Foundation (Boston, US) and Sotio a.s. (Prague, Czech Republic).

## Author contributions

A. Buqué and N.B. performed the majority of experimental procedures, collected results, analyzed data, wrote technical sections of the manuscript, and prepared figures for publication. M.P.-L., K.I., J.H., and F. Aranda provided major support to tumor establishment, survival studies and sample collection. M.P.-L. helped with preparation of samples for RNAseq and metabolomics. M.P.-L., J.G.P., S.L., L.M., and J.F. performed flow cytometry studies and analyzed data. T.Y. performed flow cytometry experiments and generated *Atg7*$^{-/-}$ TSA cells. A.S. performed biochemical and cellular studies on autophagy-competent and—deficient cells, as well as immunofluorescence microscopy assays for ER and VIM expression. S.D. performed metabolic studies. A. Boissonnas tested the effect of NAM on MMTV-PyMT mice. L.S. provided technical support to model establishment. D.E., M.H., M.K., and G.S. performed statistical and bioinformatic analyses on RNAseq data. Y.H. analyzed single-cell RNAseq data. C.M. performed immunofluorescence microscopy assays for immune infiltration under supervision from B.S. S.C.F., R.S., F. André, and S.D. provided infrastructure and guided translational/clinical relevance. O.E. supervised the analysis of single-cell RNAseq and provided infrastructure. L.Z. conceived the project and provided infrastructure and guided translational/clinical relevance. L.G. and G.K. conceived and directed the project and analyses, designed data collection protocols, designed and conducted the analyses, interpreted the results, wrote the manuscript and supervised figure preparation.

## Competing interests

S.C.F. reports funding for clinical trials from Bristol Myers Squibb, Merck and Varian, and speaker and/or advisory honoraria from Astra Zeneca, Bayer, Bristol Myers Squibb, Eisai, Elekta, EMD Serono/Merck, GlaxoSmithKline, Janssen, MedImmune, Merck US, Regeneron, Varian, and ViewRay. O.E. reports research funding from Eli Lilly, Janssen, Sanofi, equities/co-founder role in Volastra Therapeutics and OneThree Biotech, and equities/advisory role in Owkin, Freenome, Genetic Intelligence and Acuamark DX. F. André reports research funding and speaker/advisory honoraria from Roche, Astra-Zeneca, Daiichi Sankyo, Pfizer, Novartis, and Eli Lilly. L.Z. reports research funding from Bristol Myers Squibb, Roche, Glaxo Smyth Kline, Lytix Pharma, Incyte, Merus, Tusk and Pileje (completed), and from Innovate Pharma, Kaleido, Transgene, Elior, Carrefour (ongoing), consulting/advisory honoraria from Transgene, EpiVax, Lytix Biopharma, and co-founder role in everImmune. S.D. reports funding for clinical trials from AstraZeneca, Roche, Bristol Myers Squibb, Pfizer, Novartis, Amgen, Sanofi, GE Healthcare, Merck Sharp & Dohme, Eli Lilly, Puma, Myriad, Orion, Genomic Health, Pierre Fabre and Servier, non-financial support from AstraZeneca, Roche and Pfizer. G.K. reports research funding from Bayer Healthcare, Genentech, Glaxo Smyth Kline, Institut Mérieux, Lytix Pharma, PharmaMar, Sotio and Vasculox, consulting/advisory honoraria from Bristol Myers Squibb Foundation France, co-founder role of everImmune, Samsara therapeutics and Therafast Bio. L.G. reports research funding from Lytix, and Phosplatin (completed), consulting/advisory honoraria from Boehringer Ingelheim, AstraZeneca, OmniSEQ, The Longevity Labs, Inzen, and the Luke Heller TECPR2 Foundation. All other authors have no conflicts of interest to declare.
