## [Peer Review File · Nature Communications]

Reviewers' Comments:

Reviewer #1:

Remarks to the Author:

Review Buqué et al. Immunosurveillance of HR+ breast cancer- prophylactic and therapeutic effects of nicotinamide.

This is a very comprehensive labwork investigating the role of nicotinamide in HR+ breast cancer mice models. It is a proof of concept study supporting the transfer to human clinical trials.

Introduction

Page 4 line 59; it should read taxane stated therapy. Anthracyclines alone are no longer standard of care

Page 4 lin6. Please use a general term; there are more than the Recurrence Score, site others as well. Their use in the US and Europe is different. As it does not add anything to the topic it could be removed. This would be more neutral.

Page 4: line 63. This is not a very convincing reference.

Page 4 line 72: The reference No 8, has not data on survival. This is a neoadjuvant study investigating the addition of Pembrolizumab to cht in HER2 negative primary breast cancer. This is no yet fully published. The only fully published neoadjuvant study but in TNBC has been published recently in Annals Oncol 2019, epub ahead by Loibl S et al.

Page 5: line 11. This is just one study/reference. Synthetic progestins like MPA are also used as endocrine therapy in late line mBC. It has multiple effects.

Page5; NAM has multiple effects. It is used in multiple ways.

Results

I would like to ask the authors if all 7 figures are needed.

Discussion

Page 18; line 364. The majority of breast cancer are HR+ (60-75%) but the minority is linked to HRT including a progestin. This should be included in the discussion section as potential limitation.

Page 20, line 421; variety of malignancies; the cited article only deals with laryngeal cancers. All cited articles deal with melanoma or squamous cell carcinoma but not breast cancer.

Page 21: line 453ff: One need to state that this needs to be proven in clinical trials. Otherwise the last sentence is misleading.

Reviewer #2:

Remarks to the Author:

The manuscript entitled "Immunosurveillance of HR+ breast cancer – Prophylactic and therapeutic effects of nicotinamide" by Buqué et al. is an extensive study with a focus on developing a hormone receptor (HR)+ murine breast cancer model (immune competent) resembling the biology of human HR+ breast cancer, and the finding that NAM can induce immunological anti-tumor responses in this model. The findings could potentially suggest testing NAM as a safe therapeutic option in human breast cancer.

The study is comprehensive and thorough, especially the immunological part. However, there are some major weaknesses that should be addressed.

Major concerns

1) The authors aim at developing a mouse model of HR+ breast cancer that mimics that of human HR+ breast cancer. Human HR+ breast cancer is a heterogeneous group with vast differences in prognosis, treatment options and immune responses. HR+ breast cancer with a high proliferation rate usually represents Luminal B molecular type tumors. Luminal B tumor models, in contrast to Luminal A mouse syngeneic tumor models (and xenograft models), are more frequent. Luminal B tumors differ from Luminal A tumors in immune response, prognosis and therapeutic responses. It is a major weakness of the manuscript that these two HR+ breast tumor models are not commented on. If the model presented in this study represents a Luminal B model (HR+Her2- prolifhigh) this needs to be taken into consideration, in the text and conclusions, throughout the whole study. The PyMT model is also a Luminal B model. AT-3 is a TNBC model generated from the PyMT model. Immune checkpoint therapy probably has different effects in Luminal A as compared to Luminal B patients, but this is not yet known and can be commented on.

2) While the immune response is carefully investigated and described in this study, the tumor biological criteria is less carefully dealt with. ER-expression-levels of the dissected tumors should be evaluated by IHC. The most typical problem with Luminal A (and B) models are that they differentiate into basal type tumors during progression, which then will differ when it comes to prognosis and therapeutic responses and immune responses. This can be checked by using antibodies towards ER and Vimentin (HR+: ER+ and Vimentin-) on the dissected tumors in both untreated and NAM-treated groups.

3) The discussion could be improved. It is long and partly feels like a reiteration of results. The fact that Figure references are added in the text makes it easier to follow, but also makes the discussion feel like a result section.

Please consider not repeating results too much but rather discuss the findings in context of already published literature. On pg 20: Please add references and discussion on potential problems and weakness with the presented mouse model and NAM-treatment results. The manuscript would benefit from a short discussion of previous published studies and pros and cons. There are studies concerning NAM and cancer cell (not immune cell) survival that are not mentioned and that should be mentioned to be impartial and to leave the reader with an unbiased reading experience. Also, previous or ongoing studies using NAM/NAMT/NAD-inhibitors could be discussed to put this study in the right scientific perspective.

4) The Figure legends are too short and leaves the reader with unanswered questions as to how the experiments have been done. Since there is a word limit for Figure legends, it is important that full Figure legends with clear explanations are provided in Supplementary files. At this stage, some panels are difficult to evaluate because of too scarce explanations in the legends (even after studying the extended material and methods part). E.g. Amount of cells transplanted, statistical test etc.

Reviewer #3:

Remarks to the Author:

This comprehensively designed and conducted series of experiments, using a range of tumor models in different mouse strains uses nutritional-based interventions to assess tumor latency and progression as well as anti-tumor immune mechanisms and efficacy of PD1 inhibitors and methotrexate with and without NAM.

A large range of interesting experiments examines the (lack of) effect of NAM on the microbiome, and the contributions of NK, T and B cells in this model, tumor vaccination, metabolomics analyses and assessment of NAM accumulation and immune cell infiltrates within tumor tissue. Nicotinamide (NAM) is shown to be the most effective intervention for increasing TFS and OS, and is shown to abrogate the pro-carcinogenic effects of high fat and high sugar diets. The findings are novel, and suggest that NAM may have a role in both preventing breast cancer as well as enhancing/enabling immunotherapies such as PD1 inhibitors, despite breast cancer's relative lack of immune responsiveness (eg, no increase in breast cancer risk in immune suppressed transplant patients). There is suggestion that the results could indicate a role for NAM in other malignancies as well.

Main comment: the amount of NAM supplied to the animals is 0.5% w/v in drinking water (5mg per mL- no experiments using lower doses are described). If average daily water intake is ~4mL, then each ~25g mouse consumes 20mg daily of NAM. This is consistent with the amounts of niacin consumed by mice in early work showing protection from UV carcinogenesis and UV immune suppression by daily doses of 14-29mg per mouse (Gensler, 1999).

Weight by weight the amounts of NAM used in the current studies would correspond to a daily dose of 60g for a 75kg human, which is not clinically achievable. Doses shown to be UV immune protective in humans range from 500mg to 1500mg daily, 1g daily was shown to reduce skin cancer development in humans, and amounts of NAM used for radiation sensitisation for laryngeal cancer are ~4.5g (Janssens, J Clin Oncol 2012). Side effects from NAM (mainly liver function derangement) tend to start at ~4-6g daily and above.

Similarly, the calcitriol doses used (3 micrograms per mouse) are well beyond clinically possible doses in human (therapeutic doses in humans by comparison are ~1 microgram daily).

NAM may have different effects on PARP and on sirtuins at high versus low doses, and it is (always) difficult to extrapolate the anti-tumor effects of extreme NAM doses in mice to the possible effects on in vivo breast cancer prevention and treatment in humans. This needs to be discussed. The manuscript describes NAM as "currently sold over the counter as a nutritional supplement [which could] delay the development of mammary carcinogenesis.." (page 5 line 90) which implies that supplemental doses may be effective clinically. This is reiterated in the Discussion (page 20 line 413) and in the concluding sentence (page 21 line 453). NAM might prove safe and effective, but the current studies use only supra-clinical dosing and this needs to be discussed.

Reviewer #4:

Remarks to the Author:

In this manuscript, authors used mouse M/D-driven tumor to understand immunosurveillance mechanism in breast cancer. Especially authors used knockout mouse models with defects in innate and adaptive immune functions to pinpoint the cellular immune mechanism related to mouse M/D-driven tumor. However, chemical carcinogenesis model harbors serious limitations in the translation of results to understand human tumors.

Major points

1. It is not clear to define that M/D-driven tumor exactly mimic human HR+ breast cancer. Histological comparison of mouse M/D-driven tumor and human HR+ BC was not properly proved to be similar in Fig. 1C. For molecular comparison between TCGA database and mouse tumor tissue in Fig. 1D, authors should show correlation of

genome-wide gene expression level between two samples. In addition, it is not clear whether authors used TCGA BRCA dataset for this figure. Method was not fully described.

2. In Fig. 1H, authors described molecular and cellular immune profiles changed in M/D-driven tumor. To show the specificity, it should be tested in ER-AF20, ERC451A and tamoxifen-treated M/D-driven tumor bearing mice.

3. Rag2^{-/-} mice showed similar pattern with wild type mouse in TFS and OS in Fig. 2C and 2D. However, M/D-driven tumor cell lines could not be implanted successfully without MPA treatment as shown in Fig. 3. We cannot exclude the possibility of interaction between MPA and tumor microenvironment.

4. What is the molecular mechanism of NAM supplement to augment the immune response?

5. Single cell RNA-seq analysis could provide powerful evidence on molecular mediator of NAM-mediated immune activation. However, those findings in Fig 8 require the histological validation using immunohistochemical staining.

6. Molecular changes in immune cells were just increased in high dose of NAM in Fig. 8. To confirm the enhanced adaptive immunity, authors need to provide molecular evidences related to T cell clonality in single cell RNA-seq analysis.

7. Authors need to provide the molecular pathway leading transcriptional changes driven by NAM treatment.

Minor points

1. Figure 1A and 1B does not deliver novel information.

2. What was the purpose of periodic treatment of antibiotics? Is it related to the conclusion?

Reviewer #5:

Remarks to the Author:

Buque et al. developed and described a novel murine model for HR+ breast cancer. The model relies upon pharmacological intervention over time to promote oncogenesis. They then go on to show that fasting as well as nicotinamide (NAM) extends life-span in this model by delaying tumor appearance. The authors show that the tumors are under and that NAM plays a role in reactivating immunosurveillance. Though the work is impressive in scope with multiple genetic models used as well as multiple treatments explored, the experimental design for the treatments are incredibly poor. NAM is delivered by water whereas another NAD⁺ precursor, nicotinamide riboside (NR) is delivered in food. Additionally, the other drugs were delivered as IP. It could be presumed that NR and NAM could work along the same pathway. Since the delivery method differed and the dosages are not at all equal in mol amounts, it is especially difficult to conclude that NAM produced effect and NR was less effective/ineffective. The differing deliveries and differing concentrations make it difficult to conclude much of anything of the effect of one drug over another. Additionally, the use of a 24 hour fast biweekly strangely appears out of nowhere with no cited literature nor reasoning given.

Because of the experimental flaws and the poor quality of the manuscript (inappropriate/erroneous citations), I recommend to reject outright.

Major Concerns:

General:

The rate-limiting enzyme in NAD⁺ biosynthesis from NAM is NAMPT (PMID: 15381699). NAMPT inhibition was once a promising pre-clinical therapeutic avenue for cancer, which failed in clinical trials (PMID: 17924057, 19789873). Presented here is a model for increased tumor detection and clearance as a result of NAM administration. In the manuscript, you present an autophagy focused mechanism of action and briefly discuss inhibition of NAD⁺ consuming enzymes. However, you did not consider if NAM directed NAD⁺ biosynthesis is required for its action. This lack of experimentation is striking given that you also tested NR, suggesting the authors considered an NAD⁺-centric model at some point. Treatment of mice with either FK866 or equivalent would test whether it is NAM or NAD⁺ which is required. This experiment is necessary to establish a possibility of how NAM works, which is currently lacking.

The dosage of NAM given in water was used to justify 5 mM NAM in cell culture experiments. This is a confusion between dosage to an animal and the actual dosage the cells/target tissue will experience. It is very unlikely that NAM circulates in blood mM amounts given that it is normally reported in tens of μ M even when supplemented. NAM concentrations in mol per volume amounts should be measured in animals treated with NAM.

It is claimed that oncogenesis in the M/Dmodel is gut microbiome independent. The antibiotics used are absorbable leaving the possibility for an antibiotic-host interaction. Non-absorbable antibiotics are required to make clear, strong claims about gut microbiome involvement or lack thereof.

The paper refers to NAD⁺ as vitamin B3; however, this is factually incorrect. Vitamin B3 forms are nicotinamide, nicotinamide riboside, nicotinic acid, and, to a much lesser extent, tryptophan. NAD⁺ is synthesized from these vitamin B3s. This must be corrected.

Line 191 – 192: You state “Multiple epidemiological studies indicate that nutritional status has a major impact on BC incidence”. However, you cite only one citation (17). In addition, the citation deals with a reduction in calories from fat. However, you decided to perform a biweekly 24-hour fast. Can you state why you chose this regimen given the citation? Also, 24-hour fast in mice is incredibly hard on the animal due to their faster metabolic rate compared to humans. Mice normally lose ~5% body weight at the end of such a fast and then recover rapidly upon refeeding. Do you have data regarding the body mass immediately after the fasts but before re-feeding? Have you attempted shorter fasts? Why was this time chosen?

Problems with Citations:

Below are a few of the citations found to be vague or unrelated to the statement the citation was supposed to support.

Line: 89: Citation 13 deals with the effects of nicotinamide riboside (NR) on mitochondrial and stem function as well as lifespan. This citation has nothing to do with nicotinamide (NAM) being a precursor to NAD+.

Line 235: Citation 21 is a review overall for autophagy and its role in cancer and neurodegeneration. It appears to have nothing to do with a role of NAM in autophagy.

Line 419: Neither citation (assuming the numbers after Fang 2016 are the PMID) reports on the effect of NAM or NR on oncogenesis but rather focus on life-span extension. If there is a particular part of a supplement or statement within these citations that you are referring to, it must be specified.

Lines 639 and 643: What does citation 46 have to do with the mass spectrometry methodology?

Metabolomics:

There is no clarity of the number of detected ionic features, the number of those features assigned identifications, and how the identifications were assigned. The methods do not provide these details likely due to a non-inclusion of the proper citation. It is unclear why statistical comparisons were made against the fasting mice rather than to non-treated controls. It is also unclear what was detected by GCMS versus UPLC. Further, it is not clear why metabolomics was performed at all. Its inclusion appears mostly decorative and does not aid in establishing a mechanism nor in better describing the model.

Assuming a need for the data for arguments within the manuscript, I have two comments about the data included in the figures. It is odd that direct metabolites of NAM such as methyl nicotinamide and its pyridine derivatives were not included. These are very well described degradation products of NAM (PMID: 17463214, 2526576, 1491034). Were they included and not detected? If not included, why?

There appears to be a suspect identification. Nicotinic acid has often been described as being completely derived from diet and is not thought to circulate in mammals. Nicotinic acid can easily be misidentified due to in source fragmentation (PMID: 30830318, 25591916) of nicotinic acid riboside. Nicotinic acid riboside, much like NR itself (PMID: 24688693 (Figure 2B)), readily fragments to its base and ribose. The authors need to determine if an ion of m/z 256.0815 co-elutes with nicotinic acid. In general, raw data or at least peak lists of non-filtered data need to be included in the supplement.

Ref: NCOMMS-19-23566

Manuscript: **Immunosurveillance of HR+ breast cancer - Prophylactic and therapeutic effects of nicotinamide**

Authors: **Buqué *et al.***

Revision date: **Mar 20th, 2020**

Corresponding author: **Lorenzo Galluzzi**

POINT-BY-POINT REPLY TO REVIEWER N° 1

Reviewer n° 1 commented

This is a very comprehensive labwork investigating the role of nicotinamide in HR+ breast cancer mice models. It is a proof of concept study supporting the transfer to human clinical trials

Our response: We thank Reviewer #1 for appreciating the extension of our experimental approach and the ultimate aim of our work (i.e., translation to the clinic).

Then raised the following specific points

1. Introduction. Page 4 line 59; it should read taxane stated therapy. Anthracyclines alone are no longer standard of care.
2. Introduction. Page 4 lin6. Please use a general term; there are more than the Recurrence Score, site others as well. Their use in the US and Europe is different. As it does not add anything to the topic is could be removed. This would be more neutral.

Our response to points #1 and #2 from Reviewer #1: As per the constructive suggestion of Reviewer #1, we have modified the introduction of the paper to properly reflect the current clinical management of patients with newly diagnosed HR⁺ breast cancer.

3. Introduction. Page 4: line 63. This is not a very convincing reference.

Our response: We fully agree with Reviewer #1 and we have modified the reference accordingly.

4. Introduction. Page 4 line 72: The reference No 8, has not data on survival. This is a neoadjuvant study investigating the addition of Pembrolizumab to cht in HER2 negative primary breast cancer. This is not yet fully published. The only fully published neoadjuvant study but in TNBC has been published recently in Annals Oncol 2019, epub ahead by Loibl S et al.

Our response: We have corrected the Introduction to avoid pointing to data on survival in the HR⁺ setting, as per the constructive suggestions of Reviewer #1.

5. Introduction. Page 5: line 11. This is just one study/reference. Synthetic progestins like MPA are also used as endocrine therapy in late line mBC. It has multiple effects.
6. Introduction. Page 5: NAM has multiple effects. It is used in multiple ways.

Our response to points #5 and #6 from Reviewer #1: We apologize for the reductive formulation of this part of the Introduction on MPA and NAM, which we believed would help the readers to focus on our study. We have now avoided the inclusion of specific information about MPA and NAM, but focused on the factual data about their chemical composition and availability.

7. Results. I would like to ask the authors if all 7 figures are needed.

Our response: We do believe that all figures are indeed required to properly report the “very comprehensive labwork” (to cite Reviewer #1 himself or herself) we embarked upon in support of our study.

8. Discussion. Page 18; line 364. The majority of breast cancer are HR+ (60-75%) but the minority is linked to HRT including a progestin. This should be included in the discussion section as potential limitation.

Our response: We are indebted to Reviewer #1 for identifying this limitation of our study, which we have now properly discussed.

9. Page 20, line 421; variety of malignancies; the cited article only deals with laryngeal cancers. All cited articles deal with melanoma or squamous cell carcinoma but not breast cancer.

Our response: We have revised this section of the discussion as per the desiderata of Reviewer #1.

10. Page 21: line 453ff: One need to state that this needs to be proven in clinical trials. Otherwise the last sentence is misleading.

Our response: We fully agree with Reviewer #1 and we have added a sentence that clarifies the need for clinical trials demonstrating efficacy in humans.

Ref: NCOMMS-19-23566

Manuscript: **Immunosurveillance of HR+ breast cancer - Prophylactic and therapeutic effects of nicotinamide**

Authors: **Buqué et al.**

Revision date: **Mar 20th, 2020**

Corresponding author: **Lorenzo Galluzzi**

POINT-BY-POINT REPLY TO REVIEWER N° 2

Reviewer n° 2 commented

The manuscript entitled "Immunosurveillance of HR+ breast cancer – Prophylactic and therapeutic effects of nicotinamide" by Buqué et al. is an extensive study with a focus on developing a hormone receptor (HR)+ murine breast cancer model (immune competent) resembling the biology of human HR+ breast cancer, and the finding that NAM can induce immunological anti-tumor responses in this model. The findings could potentially suggest testing NAM as a safe therapeutic option in human breast cancer.

The study is comprehensive and thorough, especially the immunological part. However, there are some major weaknesses that should be addressed.

Our response: We are delighted to read that Reviewer #2 agrees with Reviewer #1 on the comprehensiveness of our study. We have done our very best to further ameliorate the study based on the constructive input from the Editors and Reviewers of *Nature Communications*, as detailed herein.

Then raised the following major points

1. 1) The authors aim at developing a mouse model of HR+ breast cancer that mimics that of human HR+ breast cancer. Human HR+ breast cancer is a heterogeneous group with vast differences in prognosis, treatment options and immune responses. HR+ breast cancer with a high proliferation rate usually represents Luminal B molecular type tumors. Luminal B tumor models, in contrast to Luminal A mouse syngeneic tumor models (and xenograft models), are more frequent. Luminal B tumors differ from Luminal A tumors in immune response, prognosis and therapeutic responses. It is a major weakness of the manuscript that these two HR+ breast tumor models are not commented on. If the model presented in this study represents a Luminal B model (HR+Her2- prolifhigh) this needs to be taken into consideration, in the text and conclusions, throughout the whole study. The PyMT model is also a Luminal B model. AT-3 is a

TNBC model generated from the PyMT model. Immune checkpoint therapy probably has different effects in Luminal A as compared to Luminal B patients, but this is not yet known and can be commented on.

Our response: We are indebted to Reviewer #2 for identifying this critical issue. We have now commented on the (known) difference between luminal A and luminal B breast tumors, and how they relate to the models we employed throughout our study.

2. While the immune response is carefully investigated and described in this study, the tumor biological criteria is less carefully dealt with. ER-expression-levels of the dissected tumors should be evaluated by IHC. The most typical problem with Luminal A (and B) models are that they differentiate into basal type tumors during progression, which then will differ when it comes to prognosis and therapeutic responses and immune responses. This can be checked by using antibodies towards ER and Vimentin (HR+: ER+ and Vimentin-) on the dissected tumors in both untreated and NAM-treated groups.

Our response: Following the suggestion of Reviewer #2, we have now evaluated ER and vimentin expression by IHC in untreated and NAM-treated tumors. We reproducibly found that MPA/DMBA-driven tumors express ER but fail to express vimentin (while the stroma expresses vimentin as it fails to express ER). We have included these data in the revised version of Figure S1D (control) and S3A (NAM treated) and discussed them in the revised version of the manuscript.

3. The discussion could be improved. It is long and partly feels like a reiteration of results. The fact that Figure references are added in the text makes it easier to follow, but also makes the discussion feel like a result section. Please consider not repeating results too much but rather discuss the findings in context of already published literature. On pg 20: Please add references and discussion on potential problems and weakness with the presented mouse model and NAM-treatment results. The manuscript would benefit from a short discussion of previous published studies and pros and cons. There are studies concerning NAM and cancer cell (not immune cell) survival that are not mentioned and that should be mentioned to be impartial and to leave the reader with an unbiased reading experience. Also, previous or ongoing studies using NAM/NAMT/NAD-inhibitors could be discussed to put this study in the right scientific perspective.

Our response: We have followed the instructions of Reviewer #2 to generate a revised Discussion section with the following features: (1) reduced length, (2) limited repetition of results, (3) discussion of potential problems and weaknesses of the model and results, (4) discussion of previous studies on NAM and cancer cell survival, (5) discussion of ongoing studies using NAM/NAMT/NAD inhibitors

4. The Figure legends are too short and leaves the reader with unanswered questions as to how the experiments have been done. Since there is a word limit for Figure legends, it is important that full Figure legends with clear explanations are provided in Supplementary files. At this stage, some panels are difficult to evaluate because of too scarce explanations in the legends (even after studying the extended material and methods part). E.g. Amount of cells transplanted, statistical test etc.

Our response: We apologize for the lack of some details in the figure legends, we have done our best to include all details in the Experimental Section, to avoid generating overlong and complex legends. Alongside, we have ensured that key details are indeed present in figure legends (e.g., statistical tests).

Ref: NCOMMS-19-23566

Manuscript: **Immunosurveillance of HR+ breast cancer - Prophylactic and therapeutic effects of nicotinamide**

Authors: **Buqué *et al.***

Revision date: **Mar 20th, 2020**

Corresponding author: **Lorenzo Galluzzi**

POINT-BY-POINT REPLY TO REVIEWER N° 3

Reviewer n° 3 commented

1. This comprehensively designed and conducted series of experiments, using a range of tumor models in different mouse strains uses nutritional-based interventions to assess tumor latency and progression as well as anti-tumor immune mechanisms and efficacy of PD1 inhibitors and methotrexate with and without NAM. A large range of interesting experiments examines the (lack of) effect of NAM on the microbiome, and the contributions of NK, T and B cells in this model, tumor vaccination, metabolomics analyses and assessment of NAM accumulation and immune cell infiltrates within tumor tissue. Nicotinamide (NAM) is shown to be the most effective intervention for increasing TFS and OS, and is shown to abrogate the pro-carcinogenic effects of high fat and high sugar diets. The findings are novel, and suggest that NAM may have a role in both preventing breast cancer as well as enhancing/enabling immunotherapies such as PD1 inhibitors, despite breast cancer's relative lack of immune responsiveness (eg, no increase in breast cancer risk in immune suppressed transplant patients). There is suggestion that the results could indicate a role for NAM in other malignancies as well.

Our response: We thank Reviewer #3 for the enthusiastic evaluation of our work, in its comprehensiveness, interest, and novelty, in thus far fully agreeing with the general evaluation from Reviewers #1 and #2.

Then raised the following major points

1. Main comment: the amount of NAM supplied to the animals is 0.5% w/v in drinking water (5mg per mL- no experiments using lower doses are described). If average daily water intake is ~4mL, then each ~25g mouse consumes 20mg daily of NAM. This is consistent with the amounts of niacin consumed by mice in early work showing protection from UV carcinogenesis and UV immune suppression by daily doses of 14-29mg per mouse (Gensler, 1999). Weight by weight the amounts of NAM used in the current studies would correspond to a daily dose of 60g for a 75kg human, which is not clinically achievable. Doses shown to be

UV immune protective in humans range from 500mg to 1500mg daily, 1g daily was shown to reduce skin cancer development in humans, and amounts of NAM used for radiation sensitisation for laryngeal cancer are~ 4.5g (Janssens, J Clin Oncol 2012). Side effects from NAM (mainly liver function derangement) tend to start at ~4-6g daily and above.

Our response: Reviewer #3 suggests that doses used in mice are not relevant for human pathophysiology based on the concentration of NAM used (0.5% w/V), a direct weight-to-weight comparison (25 g mouse versus 75 Kg human), and a daily water intake of ~4 mL. The maths provided by Reviewer #3 are correct, perhaps with the only point that water intakes in the range of 2-2.5 mL/day/mouse (in female C57BL/6 mice) have also been reported (<https://ntrs.nasa.gov/archive/nasa/casi.ntrs.nasa.gov/20160002098.pdf>), which would bring the weight-to-weight NAM comparison down to 30-40g for a 75 Kg human. Irrespectively, it is widely accepted that direct weight-to-weight comparisons between mice (or rats) and humans cannot be performed as: (1) lifespan and metabolic rates are not comparable, (2) the surface-to-weight ratio is not comparable, and (3) the intake-excretion ratio is not comparable, nor is the intake-body weight ratio. As a standalone example, mice drink 2-4 mL per day (0.08-0.16 mL/g) and urinate 0.5-1 mL per day (0.02-0.04 mL/g) (see <http://web.jhu.edu/animalcare/procedures/mouse.html>), which makes a rough 25% ratio, while young, healthy women with a normal body weight are recommended to drink an average of 2L per day (0.02 mL/g) and urinate 800-2000 mL (0.01-0.02 mL/g), at least according to multiple medical centers across the US (<http://pennstatehershey.adam.com/content.aspx?productId=117&pid=1&gid=003425>, <https://www.ucsfbenioffchildrens.org/tests/003425.html>). This makes a rough 40-100% ratio. Choosing 50% for the ease of calculations (and a similar NAM metabolism between mice and humans), women would excrete 2X the amount of NAM as compared to mice, de facto experiencing 2X lower circulating bioavailability. That said, we believe that none of the comparison above can be used as an argument in favor or against the relevance of dose, and that only clinical trials with the dose that can be administered to humans will solve this issue. Irrespectively, we measured the concentration of NAM in the circulation of NAM-treated animals, finding an average value of 16.3 µg/mL, which compares favorably with peak concentration of NAM observed in healthy volunteers upon oral administration of only 2 g NAM (which is around 30 µg/mL, see PMID 17463214). We have included these findings in the revised version of Figure S1G and discussed them in the revised version of the manuscript.

2. Similarly, the calcitriol doses used (3 micrograms per mouse) are well beyond clinically possible doses in human (therapeutic doses in humans by comparison are ~1 microgram daily).

Our response: See our response to Point #1 raised by Reviewer #3. In addition, calcitriol is entirely lateral to the main message of our story and was tested only for exploratory purposes.

3. NAM may have different effects on PARP and on sirtuins at high versus low doses, and it is (always) difficult to extrapolate the anti-tumor effects of extreme NAM doses in mice to the possible effects on in vivo breast cancer prevention and treatment in humans. This needs to be discussed. The manuscript describes NAM as “currently sold over the counter as a nutritional supplement [which could] delay the development of mammary carcinogenesis.” (page 5 line 90) which implies that supplemental doses may be effective clinically. This is reiterated in the Discussion (page 20 line 413) and in the concluding sentence (page 21 line 453). NAM might prove safe and effective, but the current studies use only supra-clinical dosing and this needs to be discussed.

Our response: We have carefully amended the discussion of our paper in view of the constructive critiques raised by Reviewer #3. In particular, we have mentioned the different effects of low- versus high-dose NAM on sirtuins and PARP, and on how the dose employed in our paper relates to humans.

Ref: NCOMMS-19-23566

Manuscript: **Immunosurveillance of HR+ breast cancer - Prophylactic and therapeutic effects of nicotinamide**

Authors: **Buqué et al.**

Revision date: **Mar 20th, 2020**

Corresponding author: **Lorenzo Galluzzi**

POINT-BY-POINT REPLY TO REVIEWER N° 4

Reviewer n° 4 commented

In this manuscript, authors used mouse M/D-driven tumor to understand immunosurveillance mechanism in breast cancer. Especially authors used knockout mouse models with defects in innate and adaptive immune functions to pinpoint the cellular immune mechanism related to mouse M/D-driven tumor. However, chemical carcinogenesis model harbors serious limitations in the translation of results to understand human tumors.

Our response: We thank Reviewer #4 for their evaluation of our paper. We have done our best to improve it based on the critiques from the Editors and Reviewers of *Nature Communications*, as detailed herein, as we discussed both the pros and cons associated with models of chemical carcinogenesis.

Then raised the following major points

1. It is not clear to define that M/D-driven tumor exactly mimic human HR+ breast cancer. Histological comparison of mouse M/D-driven tumor and human HR+ BC was not properly proved to be similar in Fig. 1C. For molecular comparison between TCGA database and mouse tumor tissue in Fig. 1D, authors should show correlation of genome-wide gene expression level between two samples. In addition, it is not clear whether authors used TCGA BRCA dataset for this figure. Method was not fully described.

Our response: We respectfully disagree on the histological similarities between human HR⁺ breast cancer and M/D-driven mammary carcinomas developing in C57BL/6 mice, which have been confirmed and commented upon by a trained pathologist (Dr. Giorgio Inghirami from Weill Cornell Medicine). As for the molecular comparisons between the transcriptional profile of M/D-driven tumors and human HR+ breast cancer, the heterogeneity of our model, which we report herein and was previously characterized from a mutational standpoint by Abbas and colleagues (Oncotarget 2015), and its human counterpart largely impedes a one-by-one comparison. Methods were provided in the dedicated section, in the following form “For comparing mouse RNAseq data with gene expression in human

breast cancer, we harnessed microarray data from breast carcinomas within the TCGA database (<http://cancergenome.nih.gov/>). We renormalized the mouse data (centered and reduced) in order to obtain similar distribution as human data. We then applied the function “subtype.cluser” of the R-package “genefu”⁵⁰ based on previously published breast cancer classification algorithms^{51,52}. Other classification methods of the package “genefu” produce similar results.”.

2. In Fig. 1H, authors described molecular and cellular immune profiles changed in M/D-driven tumor. To show the specificity, it should be tested in ER-AF20, ERC451A and tamoxifen-treated M/D-driven tumor bearing mice.

Our response: We fully agree that characterizing the molecular and cellular immune profiles of M/D-driven tumors developing in variety of different mouse models would be interesting, and we are indeed planning to pursue this possibility in the near future. That said, a deep molecular and cellular investigation of the tumor in 3 different backgrounds linked to differential (inhibited or hyperactive) ER signaling, as proposed by Reviewer #4, appears out of the scope of the current paper, which focuses on the ability of NAM to mediate prophylactic and therapeutic effects against multiple models of ER-competent breast cancer, which is the clinical entity we are modeling in this setting. In view of these observations, and in full consideration of the three R’s guidelines for animal experimentation (Replacement, Reduction and Refinement), which call for the use of the minimal number of animals, we ultimately decided not to perform these experiments as we maintained our focus on the oncopreventive and therapeutic effects of nicotinamide in ER-WT settings.

3. Rag2^{-/-} mice showed similar pattern with wild type mouse in TFS and OS in Fig. 2C and 2D. However, M/D-driven tumor cell lines could not be implanted successfully without MPA treatment as shown in Fig. 3. We cannot exclude the possibility of interaction between MPA and tumor microenvironment.

Our response: Reviewer #4 is right in saying that an interaction between MPA and the tumor microenvironment cannot be excluded. However, Fig. 3 demonstrates that M/D-driven tumors could not be transplanted from WT mice (primary hosts, where tumors were developed) to WT mice (secondary hosts, where cell lines were inoculated) irrespective of the potential interaction between MPA and the microenvironment (with a single exception that is statistically not significant). This demonstrates that the immune system of the secondary host is able to reject tumors,

most likely because of the absence of immunoediting (and hence excessive antigenicity) in the primary hosts. We have discussed this in the revised version of the paper.

4. What is the molecular mechanism of NAM supplement to augment the immune response?

Our response: As demonstrated in Figures 6-8, we believe that NAM mediate immunostimulatory effects by a variety of mechanisms, including (but potentially not limited to), (1) a direct effect on T cells and NK cells, resulting in improved cytotoxicity, (2) a direct effect on cancer cells, resulting in increased secretion of immunomodulatory and chemotactic cytokines (type I IFN, CCL2), and (3) ultimately the global repolarization of the tumor microenvironment toward a T_H1 profile with type I IFN signaling, increased antigen presentation by monocytes and improved effector functions.

5. Single cell RNA-seq analysis could provide powerful evidence on molecular mediator of NAM-mediated immune activation. However, those findings in Fig 8 require the histological validation using immunohistochemical staining.

Our response: We are indebted with Reviewer #4 for their constructive critique. As the majority of our available samples were frozen, while only a limited amount was fixed and paraffinized, we chose to use IF for CD8 T cells only and RT-PCR to investigate immune effector functions in control versus NAM-treated tumors. Largely in line with our single-cell and bulk RNA-seq data we detected a limited, but statistically significant, increase in tumor infiltration by CD8⁺ T cells, which was accompanied by increased levels of *GZMB*, *IFNG* and *PRF1* transcripts. These findings have been added to the revised version of Figure S7 and commented in the revised version of the manuscript.

6. Molecular changes in immune cells were just increased in high dose of NAM in Fig. 8. To confirm the enhanced adaptive immunity, authors need to provide molecular evidences related to T cell clonality in single cell RNA-seq analysis.

Our response: We agree with Reviewer #4 that T cell clonality may be interesting in some contexts. However, several preclinical and clinical studies fail to correlate responses to treatment with a precise variation in clonality. As

an example, in PMID 30397353, authors report that responses were associated with either a decrease OR an increase in clonality. This paper and others limit the enthusiasm about using clonality as a proxy for adaptive immune activation. To circumvent for this problem, we used flow cytometry (Fig 6F), single-cell RNAseq (Fig. 8) IF (Fig. S7) and RT-PCR (Fig. S7) to demonstrate that NAM induces adaptive immunity. In further support of this notion, the activity of NAM was abolished by CD4- and CD8- T cell depletion (Fig. 7A), providing a clear mechanistic link between NAM and adaptive immunity.

7. Authors need to provide the molecular pathway leading transcriptional changes driven by NAM treatment.

Our response: NAM is a pleiotropic molecule that influences a large variety of cellular processes, either directly or upon transformation to NAD^+ , which – unfortunately – complicates quite considerably the identification of a precise molecular pathway (admitting that a single one is indeed involved). NAD is required for life, implying that there are no means to interrupt pharmacologically or genetically NAD metabolism to determine whether NAD is responsible for the transcriptional changes imposed by NAM (because cells will die). We have demonstrated that autophagy in cancer cells is NOT responsible for the therapeutic effects of NAM. To provide further insights into this, we have now tested the ability of NAM to mediate therapeutic effects in the presence of a NAMPT inhibitor as well as of an inhibitor of the plasma membrane NAM receptor HCAR (see also our response to point #1 raised by Reviewer #5). We found that the NAMPT inhibitor FK866 *per se* limits the growth of TSA tumors developing in immunocompetent, syngeneic hosts, which is incompatible with testing its ability to modulate the therapeutic effects of NAM but supports the notion that increased NAM availability at the systemic level (as a consequence of NAMPT inhibition) is beneficial. We added these results to the revised version of Fig. S5H and discussed them in the manuscript. Conversely, the HCAR inhibitor mepenzolate bromide had no effects *per se*, but prevented NAM from mediating therapeutic effects in the same model. Moreover, it blocked the ability of NAM to drive IFN secretion by cancer cells. These data have been added to the revised version of Fig. 7D and 8F and discussed in the revised version of the paper.

As well as these minor issues

8. Figure 1A and 1B does not deliver novel information.

Our response: Figure 1A and B illustrate the experimental layout and the development of the model in immunocompetent C57BL/6 mice. As the model is highly heterogeneous, we reasoned it would be relevant to show baseline information. We would preserve Figure 1A and 1B as such.

9. What was the purpose of periodic treatment of antibiotics? Is it related to the conclusion?

Our response: No longer relevant, all antibiotic-related experiments have been removed from the manuscript, as per the request of Reviewer #5. See also our response to point #3 from Reviewer #5.

Ref: NCOMMS-19-23566

Manuscript: **Immunosurveillance of HR+ breast cancer - Prophylactic and therapeutic effects of nicotinamide**

Authors: **Buqué *et al.***

Revision date: **Mar 20th, 2020**

Corresponding author: **Lorenzo Galluzzi**

POINT-BY-POINT REPLY TO REVIEWER N° 5

Reviewer n° 5 commented

1. Buque et al. developed and described a novel murine model for HR+ breast cancer. The model relies upon pharmacological intervention over time to promote oncogenesis. They then go on to show that fasting as well as nicotinamide (NAM) extends life-span in this model by delaying tumor appearance. The authors show that the tumors are under and that NAM plays a role in reactivating immunosurveillance. Though the work is impressive in scope with multiple genetic models used as well as multiple treatments explored, the experimental design for the treatments are incredibly poor. NAM is delivered by water whereas another NAD⁺ precursor, nicotinamide riboside (NR) is delivered in food. Additionally, the other drugs were delivered as IP. It could be presumed that NR and NAM could work along the same pathway. Since the delivery method differed and the dosages are not at all equal in mol amounts, it is especially difficult to conclude that NAM produced effect and NR was less effective/ineffective. The differing deliveries and differing concentrations make it difficult to conclude much of anything of the effect of one drug over another. Additionally, the use of a 24 hour fast biweekly strangely appears out of nowhere with no cited literature nor reasoning given. Because of the experimental flaws and the poor quality of the manuscript (inappropriate/erroneous citations), I recommend to reject outright

Our response: We are surprised to read that Reviewer #5 completely dissents from all the 4 other Reviewers of Nature Communications assigned to our manuscript, pointing to the existence of “experimental flaws” and to an overall “poor quality of the manuscript”. This stands very much as odds with the global evaluations of each of the other 4 Reviewers, who classified our paper as “a very comprehensive labwork”, “comprehensive and thorough”, “comprehensively designed and conducted” and “large range of interesting experiments”. That said, we have done our best to address the points raised by Reviewer #5, as detailed here below.

Then raised the following major concerns

1. The rate-limiting enzyme in NAD⁺ biosynthesis from NAM is NAMPT (PMID: 15381699). NAMPT inhibition was once a promising pre-clinical therapeutic avenue for cancer, which failed in clinical trials (PMID: 17924057, 19789873). Presented here is a model for increased tumor detection and clearance as a result of NAM administration. In the manuscript, you present an autophagy focused mechanism of action and briefly discuss inhibition of NAD⁺ consuming enzymes. However, you did not consider if NAM directed NAD⁺ biosynthesis is required for its action. This lack of experimentation is striking given that you also tested NR, suggesting the authors considered an NAD⁺-centric model at some point. Treatment of mice with either FK866 or equivalent would test whether it is NAM or NAD⁺ which is required. This experiment is necessary to establish a possibility of how NAM works, which is currently lacking.

Our response: Reviewer #5 states that we present “an autophagy focused mechanism of action” when we instead experimentally excluded autophagy in cancer cells from both the preventive and therapeutic activity of NAM (Fig. S3B-C, Fig. S4D-E). That said, to satisfy the desiderata of Reviewer #5, we tested the therapeutic activity of NAM in the presence of FK866. We found that the NAMPT inhibitor FK866 *per se* inhibits the growth of TSA tumors developing in immunocompetent, syngeneic hosts, which is incompatible with testing its ability to modulate the therapeutic effects of NAM, but supports the notion that increased NAM availability at the systemic level (as a consequence of NAMPT inhibition) is beneficial. We added these results to the revised version of Fig. S5H and discussed them in the manuscript.

2. The dosage of NAM given in water was used to justify 5 mM NAM in cell culture experiments. This is a confusion between dosage to an animal and the actual dosage the cells/target tissue will experience. It is very unlikely that NAM circulates in blood mM amounts given that it is normally reported in tens of μ M even when supplemented. NAM concentrations in mol per volume amounts should be measured in animals treated with NAM.

Our response: We have measured NAM concentrations in the circulation of NAM-treated animals, inspired by the suggestions from Reviewer #3 and #5, finding an average value of 16.3 μ g/mL, which compares favorably with peak concentrations of NAM observed in healthy volunteers upon oral administration of 2g NAM (which is around 30 μ g/mL, see PMID 17463214). We have included these findings in the revised version of Figure S1G and discussed them in the revised version of the manuscript. See also our response to Main Comment #1 by Reviewer #3.

3. It is claimed that oncogenesis in the M/D-model is gut microbiome independent. The antibiotics used are absorbable leaving the possibility for an antibiotic-host interaction. Non-absorbable antibiotics are required to make clear, strong claims about gut microbiome involvement or lack thereof.

Our response: Although we are convinced that non-absorbable antibiotics are also very prone to interact with the host in a microbiome-independent manner, we fully agree with Reviewer #5 on the fact that our data cannot be used to make strong claims in this respect. Since they are largely peripheral to the main message of the paper, we decided to remove them.

4. The paper refers to NAD⁺ as vitamin B3; however, this is factually incorrect. Vitamin B3 forms are nicotinamide, nicotinamide riboside, nicotinic acid, and, to a much lesser extent, tryptophan. NAD⁺ is synthesized from these vitamin B3s. This must be corrected.

Our response: We have corrected this throughout the paper and we thank Reviewer #5 for spotting this imprecision. We have now systematically referred to NAM as a variant of vitamin B3 or a NAD⁺ precursor.

5. Line 191 – 192: You state “Multiple epidemiological studies indicate that nutritional status has a major impact on BC incidence”. However, you cite only one citation (17). In addition, the citation deals with a reduction in calories from fat. However, you decided to perform a biweekly 24-hour fast. Can you state why you chose this regimen given the citation? Also, 24-hour fast in mice is incredibly hard on the animal due to their faster metabolic rate compared to humans. Mice normally lose ~5% body weight at the end of such a fast and then recover rapidly upon refeeding. Do you have data regarding the body mass immediately after the fasts but before re-refeeding? Have you attempted shorter fasts? Why was this time chosen?.

Our response: We apologize for not clarifying why we choose a biweekly 24h fasting protocol. We started from published data on a single 48h-long weekly fast, which we employed in the past for subcutaneous tumors (see Pietrocola et al, Cancer Cell 2016), but reasoned that this would be too hard in a preventive setting (nearly 300 days, see F2A) and decided to split it into 2 24h-long fasting periods. We observed a consistent, rapidly recoverable 7-10% weight loss after each fasting period, as reported in Fig. 1 of this rebuttal letter, but no long-term weight loss (Fig. S2A of the paper).

Fig. 1. Weight of female C57BL/6 mice subjected to biweekly 24h fasting, as measured immediately before removing and replacing food over 4 weeks.

6. Problems with Citations. Below are a few of the citations found to be vague or unrelated to the statement the citation was supposed to support. Line: 89: Citation 13 deals with the effects of nicotinamide riboside (NR) on mitochondrial and stem function as well as lifespan. This citation has nothing to do with nicotinamide (NAM) being a precursor to NAD⁺. Line 235: Citation 21 is a review overall for autophagy and its role in cancer and neurodegeneration. It appears to have nothing to do with a role of NAM in autophagy. Line 419: Neither citation (assuming the numbers after Fang 2016 are the PMID) reports on the effect of NAM or NR on oncogenesis but rather focus on life-span extension. If there is a particular part of a supplement or statement within these citations that you are referring to, it must be specified. Lines 639 and 643: What does citation 46 have to do with the mass spectrometry methodology.

Our response: We apologize for the problems with citations, which we have systematically corrected. We thank Reviewer #5 for spotting them out.

7. Metabolomics. There is no clarity of the number of detected ionic features, the number of those features assigned identifications, and how the identifications were assigned. The methods do not provide these details likely due to a non-inclusion of the proper citation. It is unclear why statistical comparisons were made against the fasting mice rather than to non-treated controls. It is also unclear what was detected by GCMS versus UPLC. Further, it is not clear why metabolomics was performed at all. Its inclusion appears mostly decorative and does not aid in establishing a mechanism nor in better describing the model

Our response: We fully agree with Reviewer #5 on this point, and hence we decided to remove metabolomic studies from the paper (with the sole exception of circulating NAM concentrations, in response to Reviewers #3 and #5). As correctly stated by Reviewer #5, metabolomics did not add to the bulk of the experimental work and was dispensable for the main message.

8. Metabolomics. Assuming a need for the data for arguments within the manuscript, I have two comments about the data included in the figures. It is odd that direct metabolites of NAM such as methyl nicotinamide and its pyridine derivatives were not included. These are very well described degradation products of NAM (PMID: 17463214, 2526576, 1491034). Were they included and not detected? If not included, why?.

Our response: No longer relevant, see our response to point #7 raised by Reviewer #5.

9. Metabolomics. There appears to be a suspect identification. Nicotinic acid has often been described as being completely derived from diet and is not thought to circulate in mammals. Nicotinic acid can easily be misidentified due to in source fragmentation (PMID: 30830318, 25591916) of nicotinic acid riboside. Nicotinic acid riboside, much like NR itself (PMID: 24688693; Figure 2B), readily fragments to its base and ribose. The authors need to determine if an ion of m/z 256.0815 co-elutes with nicotinic acid. In general, raw data or at least peak lists of non-filtered data need to be included in the supplement.

Our response: No longer relevant, see our response to point #7 raised by Reviewer #5.

END

Reviewers' Comments:

Reviewer #2:

Remarks to the Author:

The authors have addressed my comments.

Reviewer #3:

Remarks to the Author:

The authors have now added some additional detail which aims to compare the NAM doses given to mice with plasma NAM levels observed in human volunteers after clinically achievable doses:

Page 10, line 208: "Of note, dietary supplementation of NAM to C57BL/6 mice resulted in circulating NAM levels of approx. 16.3 $\mu\text{g mL}^{-1}$ (Fig. S1G), which compare well with peak NAM concentrations observed in healthy volunteers receiving 2 g NAM per os (approx. 30 $\mu\text{g mL}^{-1}$) reference Menon et al 2007."

The Menon study volunteers received 2g of niacin (not NAM) and the peak NAM concentrations observed were less than 2.5 micrograms/mL and not the 30 micrograms/mL reported for peak niacin plasma concentration. This should be corrected in the revised manuscript.

Page 19 line 408: "The dose of NAM employed in this study (0.5% w/w with the drinking water) admittedly results in supraphysiological levels of blood borne NAM that may be difficult to achieve steadily in humans (16.3 $\mu\text{g mL}^{-1}$), but compare well with peak NAM concentrations observed in healthy volunteers receiving 2 g NAM per os (approx. 30 $\mu\text{g mL}^{-1}$). Importantly, NAM in similar dosages has been successfully used in randomized clinical trials to boost the efficacy of radiation therapy in patients with bladder and laryngeal carcinoma,^{38,39} as well as for the prevention of actinic keratosis and non-melanoma skin cancers⁴⁰,"

Again the implication that doses of NAM used in humans in previous studies of other cancers are comparable with the current murine studies (either in absolute delivered dose or in plasma concentration) should be corrected. Better to say simply that studies in humans would be needed to see whether the supraphysiological doses that (are often) used in animal studies do or don't yield comparable results.

Reviewer #4:

Remarks to the Author:

In TCGA-BRCA, 1,097 RNA-seq data are available for the comparison of mouse RNA-seq data in this manuscript. Authors used BRCA microarray data in TCGA, which might require special normalization tool for RNA-seq and microarray. Although the genetic context between mouse to human is quite different, overall expression pattern of M/D-driven mouse breast cancer and human luminal B breast cancer can be shown in parallel.

Reviewer #5:

Remarks to the Author:

Buque et al. describe a novel model for hormone receptor positive breast cancer in the mouse. Nicotinamide is then identified as a treatment and prevention within the model as a proof of principle. I appreciate the authors' responses to my comments and believe the document has greatly improved. The major strength of the manuscript is the development and validation of the pharmacologically induced model. The weaknesses surround the mechanism of action of nicotinamide. These remaining issues require textual and experimental clarification. They are described in descending importance below.

The HCAR2 Mechanism

The authors describe HCAR2 as a pharmacological target of nicotinamide. However, another name for the receptor is niacin receptor 1. Niacin, also known as nicotinic acid and nicotinate, is the pharmacological ligand for the receptor at an EC₅₀ of ~0.1 μM . However, nicotinamide, also known as niacinamide, is not a high potency ligand (doi: 10.1074/jbc.M210695200). In the larger body of the literature on the subject, nicotinamide is described as a non-agonist. In fact in citation 31 of this manuscript, on page 1174 of that citation, Elangovan et al. state "Though niacinamide is not an agonist for GPR109A, it could be converted into niacin by hydrolysis and activate GPR109A when the cells are cultured for several days as in the colony formation assay." The hydrolysis they refer to appears to be chemical, not enzymatic. Mammalian metabolism does not hydrolyze nicotinamide to nicotinic acid; though bacterial metabolism, including that in the gut microbiome, can hydrolyze the carboxamide to a carboxylic acid (doi: 10.1016/j.cmet.2020.02.001). However, the liver converts nearly all of this nicotinic acid to nicotinamide (doi: 10.1016/j.cmet.2020.02.001; doi: 10.1016/j.cmet.2018.03.018) rendering circulating nicotinic acid low to undetectable. The problem with the mechanism is further complicated by findings in one of the three original

papers describing nicotinic acid activates HCAR2. In Wise et al. 2003 (doi: 10.1074/jbc.M210695200), table 2, the IC50 for nicotinamide to remove nicotinic acid is ~100 µM. In this same article, the authors discuss that the displacement by nicotinamide could be due to a nicotinic acid impurity. This is a very reasonable explanation for the agonist activity of nicotinamide given that the preparation would need to be >99.9% pure.

These issues raise great concerns over the mechanism and findings regarding nicotinamide in this manuscript. The measured amount of nicotinamide is within the range stated in Wise et al. 2003. But it remains unclear to anyone that nicotinamide itself activates HCAR2 in vivo. It is clear though that statements such as "(MPB), an inhibitor of the plasma membrane Nam receptor...(HCAR2)" are not supported by the literature nor in this manuscript. It is possible that nicotinic acid is what was observed previously in this study as up-regulated. If this information were reintroduced the identity of nicotinic acid would still require validation with bona fide standard. Further, without quantitative measurement, the physiologic role of the possibly detected nicotinic acid is unknown. Without knowing the purity of nicotinamide, it is difficult to know if you are supplying nicotinamide or a mixture of it and nicotinic acid. Additionally, it is unknown how much nicotinamide has converted to nicotinic acid in the water given to the mice through chemical hydrolysis.

The MPB experiment is then a positive result with a poor explanation. HCAR2 may be involved, but it may be that nicotinic acid is the active molecule. This leaves open that nicotinamide should have been compared to nicotinic acid. If HCAR2 is involved in the mechanism of action, it is possible and highly predicted that nicotinic acid would be more efficacious.

Together, the role of HCAR2 is discussed in a highly inaccurate and surprising manner that requires further experimentation to be supported.

FK866 leading to increased nicotinamide rather than decreasing NAD+

In lines 307 and 308, it is stated that FK866 is expected to increase nicotinamide. The citation is then a review, which explains the role of NAMPT in NAD+ biosynthesis. It is inappropriate to assume that nicotinamide increased without measurement in blood, tissue, or urine. In fact, FK866 may decrease intracellular nicotinamide (doi:10.1016/j.molmet.2017.05.011, table 1). Without experimental evidence of NAD+ and nicotinamide within the target tissue, these conclusions are unsupported. However, even if FK866 increases nicotinamide, the well-known action of FK866 is to deplete NAD+. Without mutual exclusivity, it is unreasonable to conclude that the effect of nicotinamide is direct rather than through NAD+.

Nam inhibition of NAD+ Consuming enzymes

In the discussion, CD38 and PARP1 are mentioned as possible mechanisms of action due to nicotinamide inhibition of said enzymes. The authors discuss that SIRT1 activation rather than inhibition may be responsible for anticancer immunity. This is in line with Chatterjee et al. (citation 46 of this manuscript). However, how nicotinamide could selectively inhibit CD38 but not SIRT1 is difficult to believe, making the model in Chatterjee et al. an inappropriate citation. Further, Touat et al. (citation 47 of this manuscript), does not necessarily support this model. Touat et al. support a model in which high PARP1 activation sensitizes lung cancer cells to FK866 due to loss of NAD+. The models are discussed in a confusing manner and require clarification. Is it that nicotinamide inhibits NAD+ consuming enzymes? Or is it that nicotinamide supports NAD+ consuming enzymatic activity, in particular SIRT1?

Ref: NCOMMS-19-23566A

Manuscript: **Immunosurveillance of HR+ breast cancer - Prophylactic and therapeutic effects of nicotinamide**

Authors: **Buqué et al.**

Revision date: **May 22nd, 2020**

Corresponding author: **Lorenzo Galluzzi**

POINT-BY-POINT REPLY TO REVIEWER N° 3

Reviewer n° 3 raised the following specific points

1. The authors have now added some additional detail which aims to compare the NAM doses given to mice with plasma NAM levels observed in human volunteers after clinically achievable doses: Page 10, line 208: “Of note, dietary supplementation of NAM to C57BL/6 mice resulted in circulating NAM levels of approx. 16.3 µg mL⁻¹ µM (Fig. S1G), which compare well with peak NAM concentrations observed in healthy volunteers receiving 2 g NAM *per os* (approx. 30 µg mL⁻¹) reference Menon et al 2007.” The Menon study volunteers received 2g of niacin (not NAM) and the peak NAM concentrations observed were less than 2.5 micrograms/mL and not the 30 micrograms/mL reported for peak niacin plasma concentration. This should be corrected in the revised manuscript.

Our response: We thank Reviewer #3 for spotting this oversight. We have now corrected these sentence to (1) provide precise information on circulating NAM levels observed in healthy volunteers receiving 2 g niacin per os, and (2) avoid alluding to the comparability of NAM doses observed in humans in previous studies are comparable with those we observed in our mice (see also our response to point #2 raised by Reviewer 3).

2. Page 19 line 408: “The dose of NAM employed in this study (0.5% w/w with the drinking water) admittedly results in supraphysiological levels of blood borne NAM that may be difficult to achieve steadily in humans (16.3 µg mL⁻¹), but compare well with peak NAM concentrations observed in healthy volunteers receiving 2 g NAM per os (approx. 30 µg/ mL)¹⁹. Importantly, NAM in similar dosages has been successfully used in in randomized clinical trials to boost the efficacy of radiation therapy in patients with bladder and laryngeal carcinoma,^{38,39} as well as for the prevention of actinic keratosis and non-melanoma skin cancers⁴⁰,” Again the implication that doses of NAM used in humans in previous studies of other cancers are comparable with the current murine studies (either in absolute delivered dose or in plasma concentration) should be corrected. Better to say simply that studies in humans would be needed to see whether the supraphysiological doses that (are often) used in animal studies do or don’t yield comparable results.

Our response: Inspired by the constructive suggestions from Reviewer #3, we have now altered the discussion to avoid proposing that NAM doses observed in humans in previous studies are comparable with the levels we observed in mice, and instead comment on the need for human studies to elucidate this possibility (see also our response to point #1 raised by Reviewer 3).

Ref: NCOMMS-19-23566A

Manuscript: **Immunosurveillance of HR+ breast cancer - Prophylactic and therapeutic effects of nicotinamide**

Authors: **Buqué *et al.***

Revision date: **May 22nd, 2020**

Corresponding author: **Lorenzo Galluzzi**

POINT-BY-POINT REPLY TO REVIEWER N° 4

Reviewer n° 4 made the following request

1. In TCGA-BRCA, 1,097 RNA-seq data are available for the comparison of mouse RNA-seq data in this manuscript. Authors used BRCA microarray data in TCGA, which might require special normalization tool for RNA-seq and microarray. Although the genetic context between mouse to human is quite different, overall expression pattern of M/D-driven mouse breast cancer and human luminal B breast cancer can be shown in parallel.

Our response: We are indebted with Reviewer #4 for bringing up this point. We have now compared the transcriptomic profile of M/D-driven tumors and luminal B breast cancer patients from the TGCA dataset (which we previously identified as the human BC subtype with maximal similarity to our model, **Fig. 1D**) and included these findings in the revised version of **Supplementary Figure 1**.

Ref: NCOMMS-19-23566A

Manuscript: **Immunosurveillance of HR+ breast cancer - Prophylactic and therapeutic effects of nicotinamide**

Authors: **Buqué et al.**

Revision date: **May 22nd, 2020**

Corresponding author: **Lorenzo Galluzzi**

POINT-BY-POINT REPLY TO REVIEWER N° 5

Reviewer n° 5 commented

1. Buque et al. describe a novel model for hormone receptor positive breast cancer in the mouse. Nicotinamide is then identified as a treatment and prevention within the model as a proof of principle. I appreciate the authors' responses to my comments and believe the document has greatly improved. The major strength of the manuscript is the development and validation of the pharmacologically induced model. The weaknesses surround the mechanism of action of nicotinamide. These remaining issues require textual and experimental clarification. They are described in descending importance below.

Our response: We are delighted to hear that Reviewer #5 believes, as all the other Reviewers of *Nature Communications* do, that our paper has been greatly improved after revision. We have done our best to ameliorate it further, driven by the constructive comments from Reviewers #3, #4 and #5 under the specific guidance of the Editor of *Nature Communications*.

Then raised the following specific points

1. The HCAR2 Mechanism. The authors describe HCAR2 as a pharmacological target of nicotinamide. However, another name for the receptor is niacin receptor 1. Niacin, also known as nicotinic acid and nicotinate, is the pharmacological ligand for the receptor at an EC50 of ~0.1 µM. However, nicotinamide, also known as niacinamide, is not a high potency ligand (doi: 10.1074/jbc.M210695200). In the larger body of the literature on the subject, nicotinamide is described as a non-agonist. In fact in citation 31 of this manuscript, on page 1174 of that citation, Elangovan et al. state "Though niacinamide is not an agonist for GPR109A, it could be converted into niacin by hydrolysis and activate GPR109A when the cells are cultured for several days as in the colony formation assay." The hydrolysis they refer to appears to be chemical, not enzymatic. Mammalian metabolism does not hydrolyze nicotinamide to nicotinic acid; though bacterial metabolism, including that in the gut microbiome, can hydrolyze the carboxamide to a carboxylic

acid (doi: 10.1016/j.cmet.2020.02.001). However, the liver converts nearly all of this nicotinic acid to nicotinamide (doi: 10.1016/j.cmet.2020.02.001; doi: 10.1016/j.cmet.2018.03.018) rendering circulating nicotinic acid low to undetectable. The problem with the mechanism is further complicated by findings in one of the three original papers describing nicotinic acid activates HCAR2. In Wise et al. 2003 (doi: 10.1074/jbc.M210695200), table 2, the IC50 for nicotinamide to remove nicotinic acid is ~100 μ M. In this same article, the authors discuss that the displacement by nicotinamide could be due to a nicotinic acid impurity. This is a very reasonable explanation for the agonist activity of nicotinamide given that the preparation would need to be >99.9% pure.

These issues raise great concerns over the mechanism and findings regarding nicotinamide in this manuscript. The measured amount of nicotinamide is within the range stated in Wise et al. 2003. But it remains unclear to anyone that nicotinamide itself activates HCAR2 in vivo. It is clear though that statements such as "(MPB), an inhibitor of the plasma membrane Nam receptor...(HCAR2)" are not supported by the literature nor in this manuscript. It is possible that nicotinic acid is what was observed previously in this study as up-regulated. If this information were reintroduced the identity of nicotinic acid would still require validation with bona fide standard. Further, without quantitative measurement, the physiologic role of the possibly detected nicotinic acid is unknown. Without knowing the purity of nicotinamide, it is difficult to know if you are supplying nicotinamide or a mixture of it and nicotinic acid. Additionally, it is unknown how much nicotinamide has converted to nicotinic acid in the water given to the mice through chemical hydrolysis

The MPB experiment is then a positive result with a poor explanation. HCAR2 may be involved, but it may be that nicotinic acid is the active molecule. This leaves open that nicotinamide should have been compared to nicotinic acid. If HCAR2 is involved in the mechanism of action, it is possible and highly predicted that nicotinic acid would be more efficacious. Together, the role of HCAR2 is discussed in a highly inaccurate and surprising manner that requires further experimentation to be supported.

2. FK866 leading to increased nicotinamide rather than decreasing NAD⁺. In lines 307 and 308, it is stated that FK866 is expected to increase nicotinamide. The citation is then a review, which explains the role of NAMPT in NAD⁺ biosynthesis. It is inappropriate to assume that nicotinamide increased without measurement in blood, tissue, or urine. In fact, FK866 may decrease intracellular nicotinamide (doi:10.1016/j.molmet.2017.05.011, table 1). Without experimental evidence of NAD⁺ and nicotinamide within the target tissue, these conclusions are unsupportable. However, even if FK866 increases

nicotinamide, the well-known action of FK866 is to deplete NAD⁺. Without mutual exclusivity, it is unreasonable to conclude that the effect of nicotinamide is direct rather than through NAD⁺.

3. Nam inhibition of NAD⁺ Consuming enzymes. In the discussion, CD38 and PARP1 are mentioned as possible mechanisms of action due to nicotinamide inhibition of said enzymes. The authors discuss that SIRT1 activation rather than inhibition may be responsible for anticancer immunity. This is in line with Chatterjee et al. (citation 46 of this manuscript). However, how nicotinamide could selectively inhibit CD38 but not SIRT1 is difficult to believe, making the model in Chatterjee et al. an inappropriate citation. Further, Touat et al. (citation 47 of this manuscript), does not necessarily support this model. Touat et al. support a model in which high PARP1 activation sensitizes lung cancer cells to FK866 due to loss of NAD⁺. The models are discussed in a confusing manner and require clarification. Is it that nicotinamide inhibits NAD⁺ consuming enzymes? Or is it that nicotinamide supports NAD⁺ consuming enzymatic activity, in particular SIRT1?

Our response to points #1-3 raised by Reviewer 5: We are indebted to Reviewer #5 for bringing up these important points. Following specific instructions from the Editor of *Nature Communications*, we have removed from the manuscript all experiments involving MBP and FK866, as we fully agree with Reviewer #5 that they are too preliminary to identify a precise mechanism of action for the immunostimulatory activities of NAM we documented. Moreover, inspired by the constructive points raised by Reviewer #5 and under the guidance of the Editor of *Nature Communications*, we have carefully rewritten the discussion of the paper that speculates on the possible mechanisms of action of NAM, along the following lines: (1) autophagy activation in cancer cells is unlikely, as per the experiments with autophagy-deficient models; (2) limited cancer cell survival as a consequence of SIRT1 inhibition (as in PMID 24137378) or HCAR2 activation (as in PMID 24371223), as per the absence of overt apoptotic effects in BC cells exposed in vitro to highly supraphysiological NAM concentrations up to 80 mM and per the 1000-fold lower affinity of HCAR2 for NAM as compared to nicotinic acid (many thanks again, for bringing this up); (3) the potential, but hitherto untested implication of SIRT1 activation, based on the implication of SIRT1 in immune responses in a variety of settings (which we exemplified by citing PMIDs 28445726 and 30514750); and, alternatively (4) the potential role of CD38 or PARP1 inhibition, both of which produce NAM as part of they enzymatic reactions (and hence are susceptible to product-based inhibition) and both of which mediate immunosuppressive effects. We apologize for the citations we originally included in this latter section, which only aimed at exemplifying the immunosuppressive effects of CD38 and PARP1, rather than supporting the possibility that NAM operates in this sense. We have now cited one review article on the immunosuppressive activity of CD38 (PMID 31214171) and to original papers on the ability of PARP inhibitors to unleash type I IFN secretion (PMIDs

30589644 and 30755715). Intriguingly, these two latter papers are directly linked to our findings demonstrating that NAM not only favors type I IFN secretion by TSA cells maintained in vitro, but also (1) elicits signatures of type I IFN signaling in immune cells infiltrating TSA tumors established in immunocompetent hosts, and (2) mediates prophylactic and therapeutic effects that depend (at least in part) on type I IFN responses. We invariably specified when our speculations have not yet been experimentally validated.

END